# FairQueue: Rethinking Prompt Learning for Fair Text-to-Image Generation

**Christopher T. H. Teo**\*
christopher_teo@mymail.sutd.edu.sg

**Milad Abdollahzadeh**\*
milad_abdollahzadeh@sutd.sg

**Xinda Ma**
xinda_ma@sutd.edu.sg

**Ngai-Man Cheung**†
ngaiman_cheung@sutd.edu.sg

Singapore University of Technology and Design (SUTD)

## Abstract

Recently, prompt learning has emerged as the state-of-the-art (SOTA) for fair text-to-image (T2I) generation. Specifically, this approach leverages readily available reference images to learn inclusive prompts for each target Sensitive Attribute (tSA), allowing for fair image generation. In this work, we first reveal that this prompt learning-based approach results in degraded sample quality. Our analysis shows that the approach's training objective–which aims to align the embedding differences of learned prompts and reference images–could be sub-optimal, resulting in distortion of the learned prompts and degraded generated images.

To further substantiate this claim, **as our major contribution,** we deep dive into the denoising subnetwork of the T2I model to track down the effect of these learned prompts by analyzing the cross-attention maps. In our analysis, we propose novel prompt switching analysis: I2H and H2I. Furthermore, we propose new quantitative characterization of cross-attention maps. Our analysis reveals abnormalities in the early denoising steps, perpetuating improper global structure that results in degradation in the generated samples. Building on insights from our analysis, we propose two ideas: (i) *Prompt Queuing* and (ii) *Attention Amplification* to address the quality issue. Extensive experimental results on a wide range of tSAs show that our proposed method outperforms SOTA approach's image generation quality, while achieving competitive fairness. More resources at Project Page.

## 1 Introduction

There has been significant progress in the quality of text-to-image (T2I) generation [1–3] resulting in increasing adoption in different applications [4–10]. With this comes concerns regarding the fairness of these T2I models and their societal impacts [11–15].

**Fair T2I Generation.** T2I models may inherit biases present in their training data. Several approaches have been proposed to mitigate these biases [16–19] (See related work in Supp). Particularly, **Inclusive T2I Generation (ITI-GEN)** [16]–the existing SOTA–suggests that fair T2I approaches based on hard prompts (HP) (*e.g.*, ``A headshot of a person with fair skin tone'') are limited by linguistic ambiguity. For example, Skin Tone is often challenging to define and interpret based on HP, resulting in sub-optimal performance. To overcome this linguistic ambiguity, ITI-GEN adopts the notion that "a picture is worth a thousand words" and leverages readily available reference

---

\*Equal Contribution
†Corresponding Author

38th Conference on Neural Information Processing Systems (NeurIPS 2024).

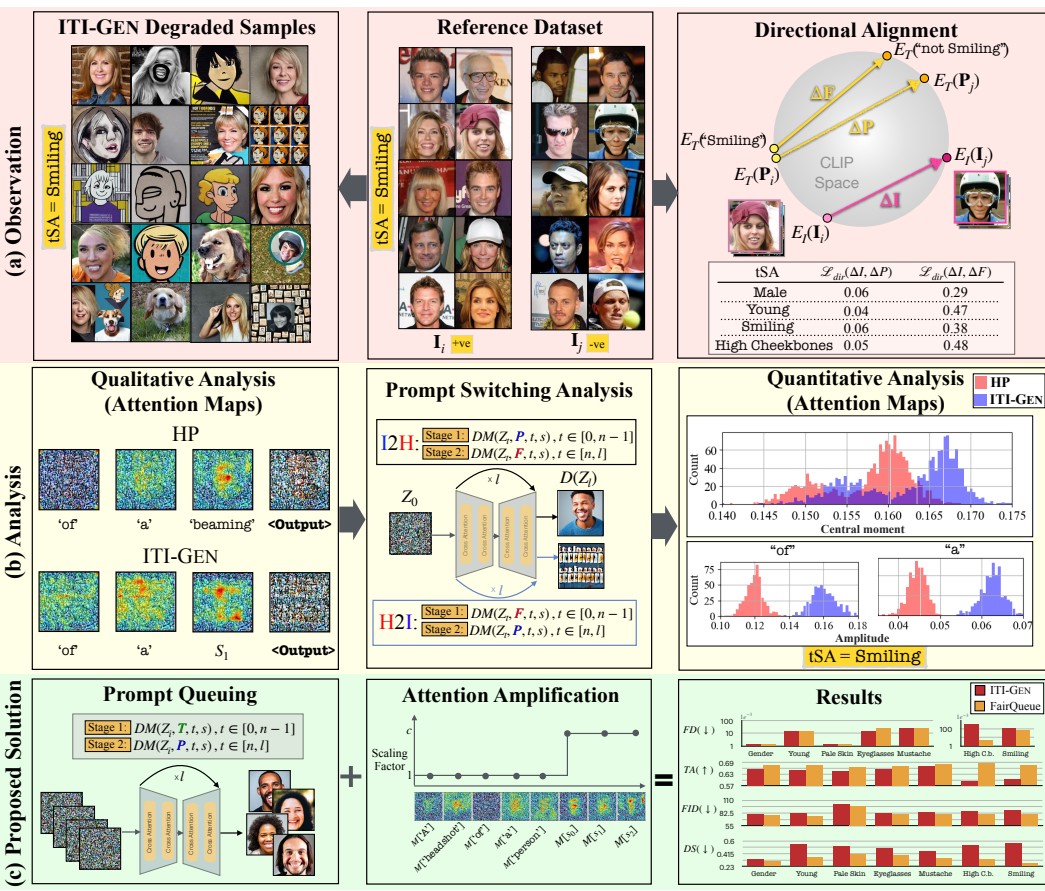

Figure 1: Our work re-visits SOTA fair T2I generation, ITI-GEN. We question ITI-GEN's central idea of prompt learning via alignment between the directions of prompt embeddings and reference image embeddings. (a) We observe degradation in images generated through ITI-GEN's learned prompts. We note that the direction of reference image embeddings could include unrelated concepts beyond tSA differences (*e.g.,* variations in accessories) resulting in learning of distorted prompts using ITI-GEN. Furthermore, we observe misalignment between the direction of credible hard prompts and that of reference images/learned prompts. (b) As our main contribution and to further understand how these distorted prompts affect the image generation process, we deep dive into the denoising network and analyze the cross-attention maps, revealing their abnormalities *e.g.,* higher activity for maps associated with non-tSA tokens (''of'', ''a''). We examine the degraded global structures resulting from these distorted prompts in the early denoising steps. Moreover, we propose I2H and H2I (Eq.2) analysis to understand impact of these degraded global structures and abnormalities in later denoising steps. In addition, we propose metrics (Eq.3) on cross-attention maps to quantify these abnormalities. (c) Building on insights from our analysis, we propose a solution to address distorted prompts while maintaining competitive fairness. Our solution FairQueue includes two ideas: prompt queuing and attention amplification. $E_T$ and $E_I$ are CLIP text and image encoder resp. [20]. $T$, $F$, $P$ are the base prompt, hard prompt with minimal linguistic ambiguity, and ITI-GEN prompt, resp.

images to learn an inclusive prompt for each tSA category. This approach translates visual attribute differences present in the reference images into prompt differences, enabling the learned prompts to be used to generate images of all tSA categories, regardless of their linguistic ambiguity. Fairness is achieved by uniformly sampling the learned prompts to condition the T2I generation. *Central to this approach is the enforcement of directional alignment between learned prompt embeddings and reference image embeddings corresponding to a pair of tSA categories.*

**In this work, we question the central idea of prompt learning via alignment between the direction of prompt embeddings and the direction of reference image embeddings in the context of fair T2I generative models.** Our work starts with examining the generated images and observes that a moderate amount of degraded images are generated based on ITI-GEN. We argue that using the direction of reference image embeddings as guidance could be sub-optimal, as the difference between reference images could include additional unrelated concepts other than the tSA difference

(Fig 1). For example, reference images of ``A headshot of a person smiling'' and ``A headshot of a person not smiling'' could contain differences in poses, accessories, hairstyling, in addition to the difference in smiling. Therefore, the direction of reference image embeddings could be noisy and include additional unrelated concepts other than the tSA difference. We perform an analysis on the direction of embeddings to further understand the issue. **We hypothesize that using the direction of reference image embeddings as guidance could lead to distortion in the learned prompts, resulting in artifacts and quality degradation in the images generated by T2I models.**

To further substantiate this claim, **as our major contribution**, we deep dive into the denoising subnetwork of the T2I model to analyze ITI-GEN prompts in the generation pipeline. Our analysis include examination of the cross-attention maps of the learned prompts at individual time steps of the denoising process. We propose novel prompt switching analysis: **ITI-GEN to HP (I2H)**, and **HP to ITI-GEN (H2I)**. We further propose new quantitative metrics for cross-attention map characterization. Our analysis reveals cross-attention maps of the learned prompts have abnormalities in the initial time steps of the denoising process. This results in synthesizing improper global structures. Interestingly, we find that the learned prompts have a minimum abnormality in the later steps–the learned prompts perform adequately in generating the desired tSA category *provided that proper global image structures could be synthesized in the initial denoising steps.* To justify our analysis of cross-attention, we remark that cross-attention contextualizes prompt embeddings with the latent representation of images and has been shown to play a key role in T2I models [21, 22].

Building on the insights of our analysis, we propose a solution to address degraded generated images without compromising fairness and diversity. Particularly, we propose Prompt Queuing to apply base prompts (without tSA tokens) in the initial time steps and ITI-GEN learned prompts in the later time steps of the denoising process. We further propose Attention Amplification to balance the quality and fairness of the T2I generation. Overall, our solution can effectively address the degraded quality issue in ITI-GEN while maintaining competitive fairness. Our contributions are:

- We examine the generated images from the prompt learning-based fair T2I generation approach and reveal a moderate amount of generated images with degradation (Sec 3.1).

- We argue that the direction of reference image embeddings could be noisy and include unrelated concepts in addition to tSA difference, and prompt learning based on alignment with the direction of reference image embeddings could be sub-optimal (Sec 3.1).

- We deep dive into the denoising subnetwork of the T2I model and analyze cross-attention maps with our proposed prompt switching analysis I2H and H2I, and our proposed quantitative metrics for cross-attention maps. Our analysis reveals and characterizes abnormalities in cross-attentions of ITI-GEN prompts in the denoising process (Sec 3.2).

- We propose FairQueue, a solution based on prompt queuing and attention amplification to improve generation quality while maintaining competitive fairness (Sec 4).

## 2   Preliminaries

**T2I Generation.** SOTA T2I generation is based on diffusion model (DM) [1–3]. In the forward diffusion process, Gaussian noise is incrementally added to the training data to train the DM. Then, during reverse diffusion, the DM generates samples by randomly sampling latent noise $Z_0 \sim N(0, \boldsymbol{I})$ as an input. For more control, text-conditioning [1, 23–25] was introduced, where we denote the reverse diffusion (denoising) of a single step $t$ by $Z_{t+1} \leftarrow DM(Z_t, \boldsymbol{R}, t, s)$. Here, $Z_t$ is the latent of the noisy image, $\boldsymbol{R}$ the input prompt, $t \in [0, l]$ the denoising step, and $s$ a random seed. Central to text conditioning is the cross-attention mechanism which contextualizes prompt embeddings with the image latent [21, 26]. Specifically the **cross-attention map $\boldsymbol{M} \in \mathbb{R}^{r \times m \times n}$**–where $r$ is the number of tokens in the prompt, and $m \times n$ shows map size for each token–is computed by:

$$\boldsymbol{M} = SoftMax(\tfrac{QK^T}{\sqrt{d}}) \tag{1}$$

where, $Q = \ell_q(\phi(Z_t))$ is the linear projection of the latent spatial features $\phi(Z_t)$, and $K = \ell_q(E_T(\boldsymbol{R}))$ is the linear projection of the textual embedding $E_T(\boldsymbol{R})$ (usually CLIP text encoder [20]). For ease of notation, we refer to the token-specific attention maps as $\boldsymbol{M}[.]$ *e.g.,* $\boldsymbol{M}[``\text{of}'']  \in \mathbb{R}^{m \times n}$ refers to the cross-attention map for the token ``of'' in $\boldsymbol{R}$. As our work focuses on the reverse diffusion process, we utilize $Z_0$ as the noisy latent input and $Z_l$ as the final latent output. This $Z_l$ is then finally passed into the DM decoder to output generated image, $D(Z_l)$.

**Fairness in Generative Models.** In generative models, fairness is defined as *equal representation* [27, 28], where for a tSA with $K$ categories, a fair generator will generate an equal number of samples for each category. As an example, for a T2I model $G$ with text prompt ``A headshot of a person'' as input, we consider $G$ as fair model *w.r.t.* tSA = Young–with two categories {Young, Old} [29, 27, 28]– if it generates an equal number of samples for each categories of this tSA [30, 31].

**Hard Prompts for Fair T2I Generation.** A baseline for achieving fairness in T2I models is to append the tSA-related prompt to the *base prompt* [17, 19]. Considering the same tSA=Young, and the base prompt ``A headshot of a person'', adding a tSA-related prompt for each category results in the HPs: ``A headshot of a person young/old''. For a fair generation, we query T2I with each of these HPs uniformly. Note that HP although very effective with certain tSA, in most cases, is ineffective due to the tSAs having linguistic ambiguity [32]–having misleading or deceptive language.

**Prompt Learning for Fair Text-to-Image Generation.** To resolve the issue of ambiguous tSAs, inspired by the recent success of prompt learning [33, 34], ITI-GEN [16] aims to achieve fairness in a pre-trained T2I model by learning inclusive tokens for each category of the tSA. Assuming the tokenized *base prompt* as $\boldsymbol{T} \in \mathbb{R}^{p \times d}$, where $p$ is the number of tokens and $d$ is the dimension of the embedding space, for each category $k \in \{1, ...K\}$ of tSA, it learns $q$ additional tokens. $\boldsymbol{S}^k = [\boldsymbol{S}_0^k, \boldsymbol{S}_1^k, \ldots, \boldsymbol{S}_{q-1}^k] \in \mathbb{R}^{q \times d}$. In [16], $q$ is set to 3. Then, *ITI-GEN prompt* is constructed by appending these learned tokens to the original tokens: $\boldsymbol{P}_k = [\boldsymbol{T}; \boldsymbol{S}^k] \in \mathbb{R}^{(p+q) \times d}$. These tokens are learned using a set of labeled reference images (*w.r.t.* tSA) $\mathcal{D}_{ref} = \{\boldsymbol{x}_i, y_i\}_{i=1}^N; y_i \in \{1, ..., K\}$ to provide stronger signals for describing tSA. More specifically, for a pair of categories $(i, j)$ of tSA, a directional loss [35] is used to match the direction of learned prompts and images for this pair in CLIP embedding [20] space *i.e.,* $\min_{\boldsymbol{S}^i, \boldsymbol{S}^j} \mathcal{L}_{dir} = 1 - \frac{(\Delta \boldsymbol{I}_{(i,j)} \cdot \Delta \boldsymbol{P}_{(i,j)})}{(|\Delta \boldsymbol{I}_{(i,j)}||\Delta \boldsymbol{P}_{(i,j)}|)}$, where $\Delta \boldsymbol{I}_{(i,j)}$ ($\Delta \boldsymbol{P}_{(i,j)}$) denotes the direction between images (text prompts) of two categories $i$ and $j$ in CLIP's embedding space, and directional loss $\mathcal{L}_{dir}$ is minimized to learn tSA tokens $\boldsymbol{S}^i, \boldsymbol{S}^j$ for these categories. Finally, using $\boldsymbol{P}_k$ as input prompt, fairness is achieved by uniformly sampling the $K$ categories of the tSA. We will omit the category index $k$ when it is clear from context, and denote learned prompt and tokens by $\boldsymbol{P}$ and $\boldsymbol{S}_0, \ldots, \boldsymbol{S}_{q-1}$ resp.

## 3 A Closer Look at Prompt Learning for Fair Text-to-Image Generation

In this section, we take a closer look at ITI-GEN [16]. First, in Sec. 3.1, we analyze ITI-GEN performance where we find quality degradation in moderate number of generated samples. We attribute this to the sub-optimal learning objective in ITI-GEN, which captures unrelated concepts that distort the learned tokens in $\boldsymbol{P}$. Then, in Sec. 3.2, we analyze ITI-GEN prompts during sample generation by inspecting the cross-attention mechanism. Our analysis reveals that ITI-GEN prompts give rise to abnormality particularly damaging to the early steps of the denoising process.

**Remark.** To conduct the following analysis on ITI-GEN prompts' behavior we require a strong baseline as a pseudo-gold standard to compare against. To address this, we found that when considering certain tSA with minimal linguistic ambiguity (MLA) [32]–a few tSA that can be described without misleading or deceptive language–HPs can serve as this strong baseline. Therefore, in this section, we focus on tSAs with minimum linguistic ambiguity. Later, in experiment section, we will include all tSAs, with or without ambiguity.

### 3.1 Limitations of Prompt Learning for Fair T2I Generation

Although ITI-GEN [16] improves fairness in T2I generation, a closer examination of its outputs reveals a potential trade-off: compromised image quality. In this section, first, we perform a systematic experiment to showcase these quality issues and then explore the potential root causes behind them.

**Experimental Setup.** To evaluate our generated samples, we utilize the metrics: i) Fairness Discrepancy (FD) [27, 31, 11, 36] to measure fairness, ii) Text-Alignment (TA) [37, 22] and FID [38] to measure quality, and iii) DreamSim (DS) [39] to measure semantic preservation. Next, we determine a set of tSA with MLA to compare ITI-GEN with HP (as a pseudo-gold standard). Specifically, we follow [16] and use pre-trained *Stable Diffusion* (SD) [1] as T2I model. Then as mentioned in Sec. 2, for HP, we append the tSA-related prompts to the base prompt. We empirically found that tSAs {Smiling, High Cheekbones}, are unambiguous by classifying 500 generated sample per HP utilizing CLIP classifier [20], where on average they both achieve a $98\%$ accuracy (Experiment details in Supp). Then, for ITI-GEN [16], we strictly follow [16] and use publicly available fair

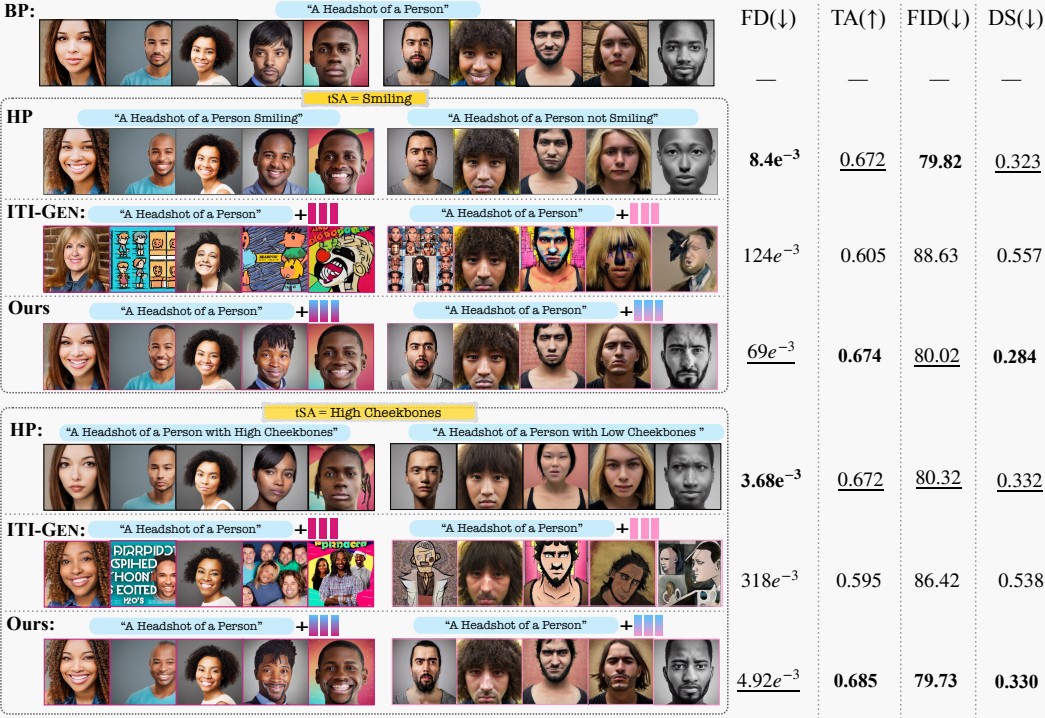

| | FD(↓) | TA(↑) | FID(↓) | DS(↓) |
|---|---|---|---|---|
| BP: | — | — | — | — |
| HP (tSA = Smiling) | **8.4e$^{-3}$** | 0.672 | 79.82 | 0.323 |
| ITI-GEN: | 124e$^{-3}$ | 0.605 | 88.63 | 0.557 |
| Ours | 69e$^{-3}$ | **0.674** | 80.02 | **0.284** |
| HP (tSA = High Cheekbones) | **3.68e$^{-3}$** | 0.672 | 80.32 | 0.332 |
| ITI-GEN: | 318e$^{-3}$ | 0.595 | 86.42 | 0.538 |
| Ours: | 4.92e$^{-3}$ | **0.685** | **79.73** | **0.330** |

Figure 2: **T2I generation performance of HP, ITI-GEN [16], and our proposed FairQueue for target Sensitive Attributes (tSAs) with minimal linguistic ambiguities.** Samples generated by HP demonstrate outstanding performance with good fairness (FD), high quality (FID and FD), and good semantic preservation (DS). Meanwhile, ITI-GEN moderately degrades sample quality, impacting fairness and semantic preservation. FairQueue demonstrates comparable performance to HP, even surpassing HP in both quality and semantic preservation in many cases. Note that HP only performs well for unambiguous tSAs, and can not be used for general fair T2I generation purposes, as it can not be defined well for ambiguous tSAs (See Supp for detailed discussion).

image dataset–sampled from CelebA [29]–as reference images to learn inclusive tokens, $S$. Finally, we generate and evaluate ITI-GEN samples based on the same latent noise input as HP. See Supp for experiment and metric details.

Fig. 2 shows some generated samples together with quantitative results. A moderate number of generated images with ITI-GEN have quality degradation often with unrelated content (*e.g.,* generating dog, multiple degraded faces, vague cartoons, etc.). Quantitative results show that for both tSAs, HP performs better in fairness (lower FD), quality (higher TA, lower FID), and semantic preservation (lower DS). We postulate degraded samples stem from ITI-GEN's sub-optimal training objective.

**Issue of Directional Loss for Fair T2I Generation.** We hypothesize that directional loss is sub-optimal in learning tSA-related tokens $S$. Particularly, the differences in reference images $\Delta I$ can include unrelated concepts in addition to variation in tSA categories. For example, considering tSA=Smiling in Fig. 1a (col 2), the reference images used for learning these two categories contain differences in pose, accessories, etc., in addition to the difference in Smiling. We further explore this potential of encoding unrelated concepts in $\Delta I$ by taking a closer look into the CLIP embedding space, where ITI-GEN's learning process happens. Recall, that as we utilize tSA with MLA and the HPs only differ in the tSA categories, we can utilize them as references in our analysis. For example, considering tSA=Smiling, the related tokenized prompts of HP in CLIP space can be computed as follows: $F_i = E_T$ ("A headshot of a person smiling"), and $F_j = E_T$ ("A headshot of a person not smiling"), with $E_T$ denoting CLIP's text encoder. Then $\Delta F = F_i - F_j$ shows the direction of the tokenized prompts in the CLIP embedding space.

Our results in Fig. 1a (col3) shows the directional loss between $\Delta I$ and $\Delta F$ *i.e.,* $\mathcal{L}_{dir}(I, F)$, for different tSAs using the reference images for each tSA. Note that $\mathcal{L}_{dir} = 0$ means perfect alignment. Our comparison reveal considerable misalignment between $\Delta I$ and $\Delta F$ implying that unrelated concepts are potentially encoded in $\Delta I$. Meanwhile, $\Delta I$ and $\Delta P$ near perfect alignment implies that

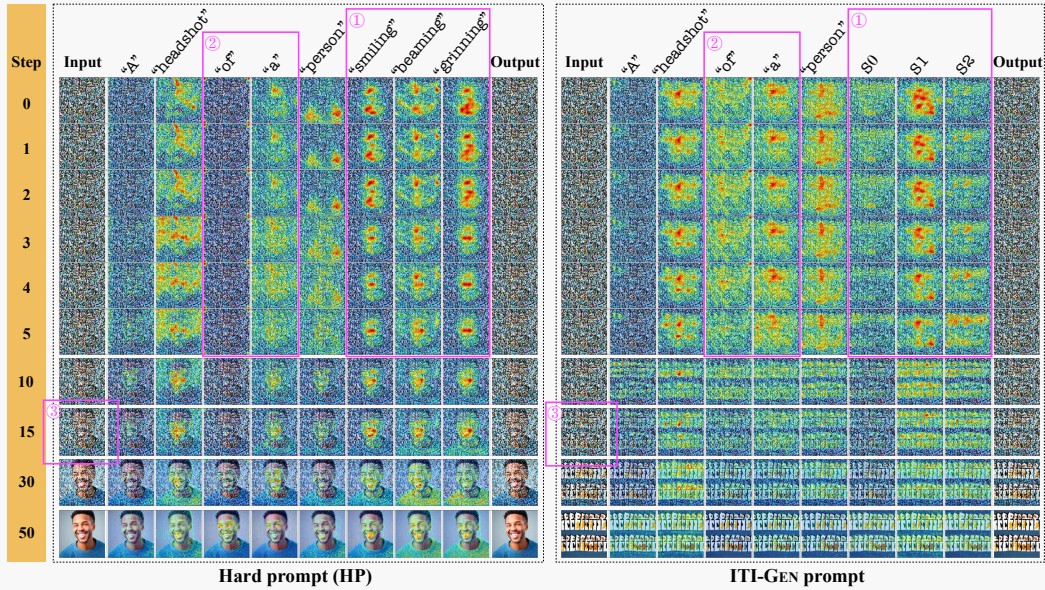

Figure 3: **Comparison of cross-attention maps during the denoising process with HP (left) and ITI-GEN (right).** Here, we use tSA=Smiling and plot the denoising process for one sample generation. Each denoising process consists of $l = 50$ steps initiated with the same noisy input. Each cell depicts the attention map for the respective token (column) at the respective step (row) overlaid on the input. We highlight 3 key observations: ① ITI-GEN tokens $S_i$ have abnormal activities compared to the corresponding tSA-related tokens in HP by attending to unrelated regions (backgrounds) or scattered attention. ② non-tSA tokens like ''of'' and ''a'' are abnormally more active in the presence of ITI-GEN tokens. ③ As compared to the HP counterpart, issues created by ITI-GEN tokens (① & ②) degrade the global structure in the early denoising steps (*e.g.*, Step 15), for example, human face in HP vs some unrelated structure in ITI-GEN. The same behavior is observed for some other samples and tSAs (see Supp for more samples, and other tSAs with more denoising steps).

these unrelated concepts are potentially transferred to $P$ via ITI-GEN's learning objective, resulting in distorted learned token $S$.

## 3.2 Analyzing the Effect of ITI-GEN Prompts in T2I Generation

In the previous section, we observed degraded sample quality in ITI-GEN which we attribute to the sub-optimal training objective that results in learning distorted tokens. In this section, we take a step further to answer the question: *"Given a pre-trained T2I model and some distorted learned prompts as input, how do these distorted prompts affect the image generation process of the T2I model?"*

To answer this, we deep dive into the latent denoising network [1] and analyze the cross-attention mechanism [40]–the bridge for text and image modules in T2I models [1, 23–25]. In this analysis, *we visualize the cross-attention maps to investigate potential anomalies caused by distorted tokens in the denoising process.* Specifically, we compare cross-attention maps of ITI-GEN prompt against HP with minimal linguistic ambiguity (as reference). To allow fair token-to-token comparison, in this experiment, we lengthen HP by including additional tokens containing synonyms of the tSA. Note that this did not augment HP's behavior, and similar results are seen in the original HP. See Supp for more details.

**Visualizing Cross-attention Maps.** We follow DAAM [41] for visualizing cross-attention maps by tracing attention scores in the cross-attention module to demonstrate how an input token within a prompt influences parts of the generated image. Specifically, to visualize the cross-attention map of a token, DAAM interpolates and accumulates the attention scores over all scales (layer of the U-Net [42] as the denoising network [1]), and all denoising steps. However, we tailor DAAM to the requirements of our fine-grained analysis by introducing further controls. First, we isolate the attention maps for each denoising step to allow for both step-wise and multi-step analysis. Second, we introduce a prompt-switching mechanism, allowing for the interchangeable tracing of different prompts at any particular denoising step.

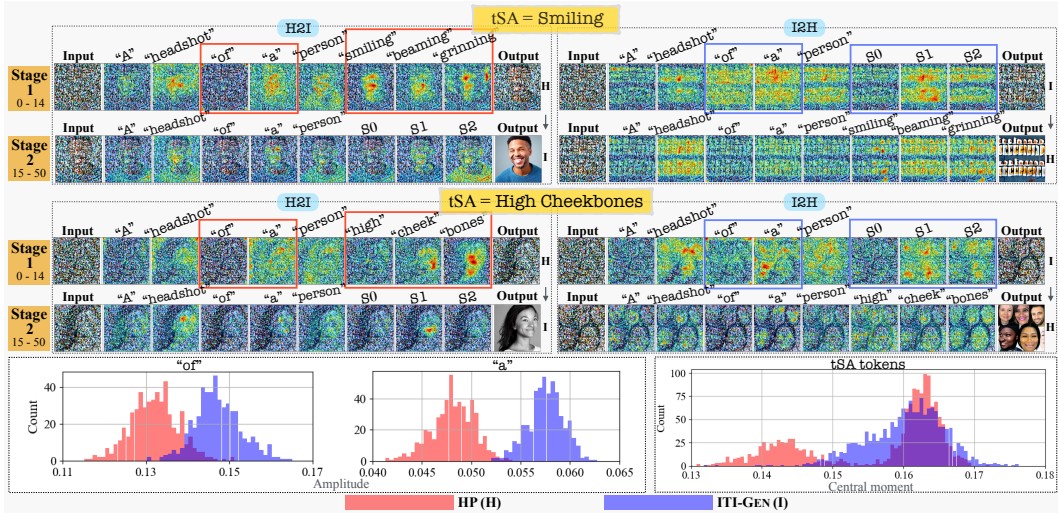

Figure 4: **Analyzing the accumulated cross-attention maps for the denoising process in our proposed prompt switching analysis I2H and H2I.** Here, we use two tSAs: `Smiling`, `High Cheekbones`. For each tSA, we show the accumulated cross-attention maps for H2I and I2H, with some quantitative results. In the H2I (I2H) experiment, the first row shows the accumulated cross-attention maps during the early denoising steps with HP (ITI-GEN) as the input prompt, and the second row shows the maps during later steps, after switching to ITI-GEN (HP). **Observation 1:** *Learned tokens in* ITI-GEN *affect early denoising steps, degrading global structure synthesis; such degraded global structure disrupts the final output.* This is observed in I2H. **Observation 2:** *Learned tokens in* ITI-GEN *works decently in the later stage of the denoising process if the global structure is synthesized properly.* This is observed in H2I. As we show in Supp, similar observations can be made for other samples and other tSAs. Bottom: Histograms of our proposed metrics on cross-attention maps demonstrate the abnormalities in many samples.

**Abnormalities in the Presence of Distorted Tokens.** To investigate potential anomalies arising from distorted tokens learned by ITI-GEN, we comprehensively analyze cross-attention maps of all denoising time steps on 500 generated samples per category of each tSA, for both ITI-GEN and HP. Fig. 3 shows one of these cross-attention maps comparing ITI-GEN and HP for tSA=`Smiling` (more examples in Supp). Our empirical investigation reveals four points: i) global structure is synthesized in the early steps of the denoising process aligning with previous works [21] that the denoising process progressively synthesizes the image. ii) Learned ITI-GEN tokens have abnormal attention compared to the tSA-related tokens in HP (''`Smiling`'' in col. 7 of HP), *e.g.,* $M[S_i]$ contain unrelated or scatter activation (**Issue 1**). iii) In the presence of the ITI-GEN tokens, other non-tSA tokens (like $M[$'`a`'$]$ and $M[$'`of`'$]$) are abnormally more active (**Issue 2**). We remark that tokens interact with each other in the denoising steps. iv) Considering **Issues 1 & 2**, we observe that degraded global structure is synthesized in the early steps of denoising, and eventually a degraded sample is generated at the end of the denoising. Note that similar issues occur with many other samples and tSAs (details in Supp).

To further understand issues, we isolate effect of distorted tokens by proposing two analyses focusing on different denoising steps. These two analyses dissect the influence of distorted tokens in key denoising steps (Recall denoising of a single step $t$ is denoted by $Z_{t+1} \leftarrow DM(Z_t, \boldsymbol{R}, t, s)$, Sec. 2):

$$\text{I2H} = \begin{cases} DM(Z_t, \boldsymbol{P}, t, s) & t \in [0, n-1] \\ DM(Z_t, \boldsymbol{F}, t, s) & t \in [n, l] \end{cases} , \text{H2I} = \begin{cases} DM(Z_t, \boldsymbol{F}, t, s) & t \in [0, n-1] \\ DM(Z_t, \boldsymbol{P}, t, s) & t \in [n, l] \end{cases} \quad (2)$$

To do this, as seen in Fig.1b (col 2), we first propose **Analysis 1: switching prompt from ITI-GEN to HP (I2H)** during the denoising process. This allows for a better understanding of how the global structure's degradation in early denoising steps may affect the final generated output. Specifically, as in Eq. 2, I2H first utilizes *ITI-GEN prompt* $\boldsymbol{P}$ in early denoising steps which potentially leads to degraded global structure. This is then followed by utilizing *hard prompt* $\boldsymbol{F}$ for the remaining denoising steps. Next, to investigate if ITI-GEN tokens will create the same issues in the later steps of the denoising process, we propose **Analysis 2: HP to ITI-GEN (H2I)**. Converse to the previous experiment, we utilize $\boldsymbol{F}$ in early steps of denoising, and then switch to using $\boldsymbol{P}$ as input prompt. For each experiment, we plot and analyze the cumulative cross-attention maps for early steps (0 to $n-1$) and later steps ($n$ to $l$) separately. Fig. 4 shows an example of the cross-attention maps for

these two experiments with tSA={Smiling, High Cheekbones}. See Supp for more samples and details. Considering results in Fig. 4 the following observations can be made:

**Observation 1:** *Learned tokens in* ITI-GEN *affect the early steps of the denoising process leading to degradation in synthesizing global structure.* More specifically comparing the first row of the cross-attention map between I2H and H2I in Fig. 4, we can have the following observations: i) ITI-GEN tokens have more scattered attention or attending to unrelated regions compared to tSA-related tokens in HP; ii) non-tSA tokens like ''a'', and ''of'' are more active in the presence of the ITI-GEN tokens. These two abnormalities result in degraded global structure in the early steps. In addition, considering the second row of the I2H in Fig. 4, the degraded global structure in the early steps leads to disrupted final output even though the (non-distorted) HP prompt is used in later steps.

**Observation 2:** *Learned tokens in* ITI-GEN *works decently in the later steps of the denoising process if the global structure is synthesized properly.* More specifically, considering H2I in Fig. 4, when HP prompts synthesize proper global structure in early steps, the ITI-GEN tokens attend to proper regions and contribute to adding the finer details related to tSA, as shown in the second row of H2I.

**Quantitative Metrics for Cross-attention Maps.** In addition to the visual demonstration, we propose two metrics to support further our observed abnormalities of ITI-GEN tokens in the early steps of denoising for a large number of generated samples. Specifically, for each generated sample: i) To quantify abnormally active attention associated with non-tSA tokens, we compute the expectation of **attention amplitude**: $\mathbb{E}_{(x,y)}\{M[J]\}$, where $J$ is a non-tSA token such as ''of''. ii) We analyze the scatter in attention by measuring the second **central moment** [43] for each tSA token $K$:

$$\mu(K) = \sum_{x,y} \{[(x - \bar{x})^2 + (y - \bar{y})^2]\tilde{M}[K]_{(x,y)}\} \tag{3}$$

Here, $\tilde{M}[K] = (M[K]/\sum_{x,y} M[K])$, and $(\bar{x}, \bar{y})$ is the centroid. The two metrics are computed on the accumulated cross-attention maps from stage 1 of I2H (for ITI-GEN) and H2I (for HP). The histograms of these two metrics for 500 generated samples in Fig. 1b (col 3) and Fig. 4 demonstrate **Issues 1 & 2** in many generated samples.

**Remark.** Our thorough analysis in this section shows that distorted tokens learned by ITI-GEN only have destructive performance in the early steps of denoising, and they generally have decent performance in later steps when the global structure is formed properly (H2I). *We remark that even though H2I has decent performance in fair and high-quality T2I generation, it is only applicable to tSA with minimal linguistic ambiguity. In the next section, we will discuss our proposed method to address fair and high-quality T2I generation encompassing both ambiguous and unambiguous tSA.*

## 4 Proposed Method

In this section, we present our proposed method, FairQueue a new generation framework consisting of two additions: *Prompt Queuing* and *Attention Amplification* to improve the sample quality when implementing fair T2I generation. In addition to quality improvements, FairQueue also allows for better semantic preservation of the original sample generated from the base prompt $T$.

**Prompt Queuing.** Recall that when utilizing ITI-GEN prompt $P$–which is tuned to generate samples containing the tSA–degraded global structure occurs in early denoising steps for a moderate number of samples. Conversely, utilizing HP with minimal linguistic ambiguity enables high-quality and fair T2I generation. However, as such HPs are not available for all tSAs [16], we naturally consider the next best available option–the base prompt $T$ (a natural language prompt without the distorted trainable tokens)–and propose prompt queuing. Specifically, as seen in Fig. 1(c) , prompt queuing first utilizes $T$ in the early $n$ denoising steps, thereby allowing for the global structures to form properly. Next, we transit to ITI-GEN prompt $P$ for the remaining $(l - n)$ steps. This allows the more fine-grained tSA semantics to be developed on top of the already well-defined global structures.

**Attention Amplification.** By implementing prompt queuing, the output samples may experience a reduction in tSA expression due to the reduced exposure to the ITI-GEN prompt $P$. To address this, we propose Attention Amplification, an intuitive solution that emphasizes the expression of the tSA by scaling the ITI-GEN token's cross-attention maps, *i.e.,* $c * M[S_i]$ where $c > 1$.

## 5 Experiments

In this section, we evaluate our proposed (FairQueue) against the existing SOTA ITI-GEN [16] over various tSA. Then, we conduct an ablation study by first evaluating the contribution brought by each

Table 1: **Evaluating Proposed FairQueue against ITI-GEN .**We utilize *FD:* Fairness Discrepancy (↓), *TA:* Text-Alignment (↑), *FID* (↓), and *DS:* DreamSim (↓) to determine the fairness, quality, and semantic preservation, respectively. For FD a combination of CLIP [20], off-the-shelf classifier [44, 45] and human evaluator were utilized as tSA classifier. For TA we utilize CLIP [20] as the feature extractor. Overall, our proposed method demonstrates the best ability to balance between sample quality and fairness, while preserving the semantics of the original base-prompt $T$.

| tSA | | FD (↓) | TA (↑) | FID (↓) | DS (↓) |
|---|---|---|---|---|---|
| **Single tSA (CelebA)** | | | | | |
| Gender | ITI-GEN | $\mathbf{6.41e^{-3} \pm 4.2e^{-3}}$ | $0.655 \pm 1.2e^{-2}$ | $78.9 \pm 1.3$ | $0.337 \pm 1.4e^{-2}$ |
| | Ours | $\mathbf{6.41e^{-3} \pm 3.8e^{-3}}$ | $\mathbf{0.676 \pm 5.2e^{-3}}$ | $\mathbf{78.3 \pm 1.5}$ | $\mathbf{0.308 \pm 1.2e^{-2}}$ |
| Young | ITI-GEN | $\mathbf{13.1e^{-3} \pm 8.1e^{-3}}$ | $0.653 \pm 9.4e^{-3}$ | $82.9 \pm 1.4$ | $0.552 \pm 3.2e^{-2}$ |
| | Ours | $15.5e^{-3} \pm 3.8e^{-3}$ | $\mathbf{0.678 \pm 8.1e^{-3}}$ | $\mathbf{75.3 \pm 2.1}$ | $\mathbf{0.370 \pm 2.7e^{-2}}$ |
| Smiling | ITI-GEN | $124e^{-3} \pm 9.2e^{-3}$ | $0.605 \pm 1.2e^{-2}$ | $88.6 \pm 0.9$ | $0.557 \pm 2.2e^{-2}$ |
| | Ours | $\mathbf{69.0e^{-3} \pm 4.2e^{-3}}$ | $\mathbf{0.674 \pm 1.7e^{-2}}$ | $\mathbf{80.0 \pm 1.3}$ | $\mathbf{0.284 \pm 1.0e^{-2}}$ |
| High Cheekbones | ITI-GEN | $318e^{-3} \pm 12.0e^{-3}$ | $0.595 \pm 1.2e^{-3}$ | $86.40 \pm 2.1$ | $0.538 \pm 1.6e^{-2}$ |
| | Ours | $\mathbf{4.92e^{-3} \pm 3.6e^{-3}}$ | $\mathbf{0.685 \pm 7.2e^{-3}}$ | $\mathbf{79.7 \pm 2.4}$ | $\mathbf{0.330 \pm 2.2e^{-2}}$ |
| Pale Skin | ITI-GEN | $\mathbf{1.41e^{-3} \pm 1.2e^{-3}}$ | $0.646 \pm 1.8e^{-2}$ | $101.3 \pm 4.6$ | $0.525 \pm 2.8e^{-2}$ |
| | Ours | $\mathbf{1.41e^{-3} \pm 1.2e^{-3}}$ | $\mathbf{0.666 \pm 1.9e^{-2}}$ | $\mathbf{97.0 \pm 3.2}$ | $\mathbf{0.408 \pm 3.0e^{-2}}$ |
| Eyeglasses | ITI-GEN | $\mathbf{14.1e^{-3} \pm 2.6e^{-3}}$ | $0.654 \pm 3.3e^{-3}$ | $83.5 \pm 1.4$ | $0.486 \pm 1.4e^{-2}$ |
| | Ours | $25.4e^{-3} \pm 1.9e^{-3}$ | $\mathbf{0.670 \pm 6.1e^{-3}}$ | $\mathbf{79.4 \pm 2.3}$ | $\mathbf{0.391 \pm 1.6e^{-2}}$ |
| Mustache | ITI-GEN | $26.2e^{-3} \pm 1.8e^{-3}$ | $0.670 \pm 4.2e^{-3}$ | $85.0 \pm 3.3$ | $0.452 \pm 1.9e^{-3}$ |
| | Ours | $\mathbf{22.6e^{-3} \pm 1.2e^{-3}}$ | $\mathbf{0.680 \pm 5.3e^{-3}}$ | $\mathbf{80.2 \pm 3.0}$ | $\mathbf{0.345 \pm 3.1e^{-3}}$ |
| Chubby | ITI-GEN | $\mathbf{112e^{-3} \pm 8.8e^{-3}}$ | $0.647 \pm 2.2e^{-3}$ | $79.2 \pm 1.5$ | $0.551 \pm 3.6e^{-3}$ |
| | Ours | $119e^{-3} \pm 7.2e^{-3}$ | $\mathbf{0.675 \pm 2.3e^{-3}}$ | $\mathbf{78.3 \pm 1.4}$ | $\mathbf{0.387 \pm 3.0e^{-3}}$ |
| Gray Hair | ITI-GEN | $286e^{-3} \pm 6.8e^{-3}$ | $0.640 \pm 4.3e^{-3}$ | $87.3 \pm 2.1$ | $0.533 \pm 2.9e^{-3}$ |
| | Ours | $\mathbf{266e^{-3} \pm 7.1e^{-3}}$ | $\mathbf{0.669 \pm 3.7e^{-3}}$ | $\mathbf{82.2 \pm 2.3}$ | $\mathbf{0.417 \pm 3.1e^{-3}}$ |
| **Multi tSA (CelebA)** | | | | | |
| Gender × Young | ITI-GEN | $39.1e^{-3} \pm 1.2e^{-3}$ | $0.668 \pm 7.1e^{-3}$ | $72.6 \pm 3.1$ | $0.458 \pm 7.8e^{-3}$ |
| | Ours | $\mathbf{12.4e^{-3} \pm 2.3e^{-3}}$ | $\mathbf{0.686 \pm 5.7e^{-3}}$ | $\mathbf{71.7 \pm 2.5}$ | $\mathbf{0.373 \pm 4.4e^{-3}}$ |
| Gender × Young × Eyeglasses | ITI-GEN | $257e^{-3} \pm 8.7e^{-3}$ | $0.654 \pm 3.3e^{-3}$ | $65.2 \pm 1.6$ | $0.475 \pm 1.1e^{-3}$ |
| | Ours | $\mathbf{208e^{-3} \pm 7.3e^{-3}}$ | $\mathbf{0.671 \pm 4.1e^{-3}}$ | $\mathbf{61.5 \pm 2.7}$ | $\mathbf{0.360 \pm 6.3e^{-3}}$ |
| Gender × Young × Eyeglasses × Smiling | ITI-GEN | $190e^{-3} \pm 1.7e^{-2}$ | $0.643 \pm 7.7e^{-3}$ | $65.5 \pm 2.7$ | $0.475 \pm 9.1e^{-3}$ |
| | Ours | $\mathbf{168e^{-3} \pm 1.0e^{-2}}$ | $\mathbf{0.661 \pm 2.4e^{-3}}$ | $\mathbf{60.8 \pm 1.1}$ | $\mathbf{0.379 \pm 9.7e^{-3}}$ |
| **Multi tSA (Fairface & Fair Benchmark)** | | | | | |
| Gender × Age | ITI-GEN | $142e^{-3} \pm 4.2e^{-3}$ | $0.659 \pm 7.2e^{-3}$ | $\mathbf{58.24 \pm 3.4}$ | $0.445 \pm 1.2e^{-3}$ |
| | Ours | $\mathbf{108e^{-3} \pm 4.3e^{-3}}$ | $\mathbf{0.672 \pm 1.1e^{-3}}$ | $58.81 \pm 3.3$ | $\mathbf{0.359 \pm 3.5e^{-3}}$ |
| Gender × Skin Tone | ITI-GEN | $166e^{-3} \pm 3.7e^{-3}$ | $0.670 \pm 2.2e^{-3}$ | $59.56 \pm 3.6$ | $0.463 \pm 7.7e^{-3}$ |
| | Ours | $\mathbf{116e^{-3} \pm 4.4e^{-3}}$ | $\mathbf{0.686 \pm 2.3e^{-3}}$ | $\mathbf{54.66 \pm 2.7}$ | $\mathbf{0.390 \pm 1.8e^{-3}}$ |

component for FairQueue *i.e.,* Prompt queuing, and Attention Scaling. Then we revisit the task initially proposed by ITI-GEN : Training Once-for-All Token. Overall, we show that FairQueue achieves new SOTA performance.

**Experimental Setup.** Following [16], we utilize the publicly available reference dataset from CelebA [29], FairFace [45] and FAIR benchmark [44]. For CelebA, we perform both single tSA and multi-tSA experiments. Note that in this dataset each tSA has two categories. In FairFace and FAIR datasets, tSAs have more categories, *e.g.,* Age and Skin tones contain 9 and 6 classes, respectively. Therefore, fair generation is more challenging in these datasets. For these experiments we use $T=E_t(\texttt{"A headshot of a person"})$ as base prompt, and for a fair comparison, we utilize exactly the same learned $P$ from ITI-GEN's original code [46] for both ITI-GEN and proposed FairQueue. In addition, we randomly sample a set of 500 latent codes and use the same latent codes for both approaches. As earlier discussed in Sec. 3, we utilize fairness discrepancy (FD) to evaluate fairness, Text-Alignment (TA) and FID for quality, and DreamSim (DS) to measure semantic preservation. See Supp for more details. We repeat this process 5 times and report the mean and standard deviation.

**Our results** in Tab. 1 demonstrate that FairQueue is able to match ITI-GEN fairness performance closely, and in some cases even improve upon it. For example, for tSA=Smiling, FairQueue indicates a significantly lower bias (FD=$6.9e^{-3}$) than ITI-GEN (FD=$124e^{-3}$). In addition, considering sample quality, FairQueue achieves an overall better performance than ITI-GEN for all datasets. For example, in CelebA, FairQueue's TA≥ 0.666 while ITI-GEN's TA≤ 0.655, with the worst performance with High Cheekbones (TA=0.595). These results are similarly reflected in FID. We remark that this quality degradation largely contributes to ITI-GEN fairness degradation. Finally, when considering semantic preservation (DS ↓) of the original sample generated with $T$, FairQueue achieves unparalleled performance by ITI-GEN.

**Ablation study: evaluating prompt queuing and attention scaling.** To evaluate the contribution brought by FairQueue , we consider the same setup as Sec. 5 in the main manuscript focusing on the tSA `Smiling`. Here, we compare the performance when utilizing different attention amplification scaling factors, $c$, and different prompt queuing transition points *i.e.,* switching from $T$ to $P$. Specifically, we consider gradual increments in $c \in [0, 12]$ and shifting of the transition points, step $\in \{0, 0.1l, 0.2l, 0.3l\}$ when $l = 50$.

Our results in Fig.5 illustrate that generally when $c$ increases, fairness improves. However, a saturation point ($c = 10$) exists where quality and semantic preservation beyond this point degrades. Then when considering different transition points, we find that at step $0.2l$ FairQueue achieves the best quality and semantic preservation performance while still achieving good fairness measurements. However, increasing beyond this point results in significant fairness degradation.

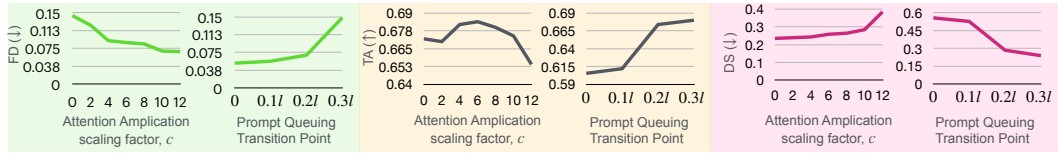

Figure 5: **Ablation Study:** Comparing FairQueue performance when varying i) attention amplification factor, $c$ or ii) Prompt Queuing transition point from $T \rightarrow P$ for tSA `Smiling`.

**Ablation study: revisiting training once-for-all token.** Utilizing FairQueue we follow [16] and re-visit adapting pre-trained ITI-GEN tokens, $S_i$ to a new Base Prompt $T' = E_t($"`A headshot of a doctor`"$)$ by pre-pending. Then we generate samples utilizing both FairQueue and ITI-GEN with the same noise input. As seen in Fig. 6 FairQueue demonstrates better performance than ITI-GEN , achieving both better quality and semantic preservation of the sample generated by $T'$ while still having good tSA representation—more illustration in Supp.

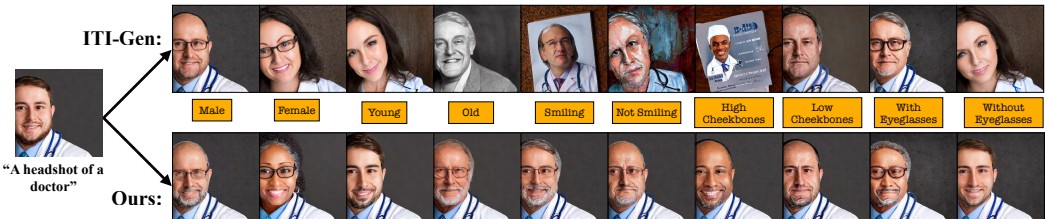

Figure 6: **Illustration of samples generated by ITI-GEN and FairQueue with a new Base Prompt $T' = E_t\{$"`A headshot of a doctor`"$\}$ via pre-pending.** FairQueue improves sample quality and ability to preserve the original sample's semantics while mainly adapting only the tSA.

## 6 Conclusion

In this paper, we reveal quality degradation in ITI-GEN –the existing SOTA fair T2I prompt learning approach. Our analysis reveals that this quality degradation is due to the distorted learned tokens in ITI-GEN prompt impacting cross-attention in the early steps of the denoising (reverse diffusion) process. To address this, we propose FairQueue a simple but effective solution consisting of: Prompt Queuing and Attention Amplification. Overall, our extensive experimentation demonstrates FairQueue achieves new SOTA performance in balancing quality, fairness, and semantic preservation. **Limitation, related work and additional experiments can be found in the Supp.**

## Acknowledgements

This research is supported by the National Research Foundation, Singapore under its AI Singapore Programmes (AISG Award No.: AISG2-TC-2022-007); The Agency for Science, Technology and Research (A*STAR) under its MTC Programmatic Funds (Grant No. M23L7b0021). This research is supported by the National Research Foundation, Singapore and Infocomm Media Development Authority under its Trust Tech Funding Initiative. Any opinions, findings and conclusions or recommendations expressed in this material are those of the author(s) and do not reflect the views of National Research Foundation, Singapore and Infocomm Media Development Authority.

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

# Supplementary

This supplementary provides additional experiments as well as details that are required to reproduce our results. These were not included in the main paper due to space limitations. The supplementary is arranged as follows:

# A More Experimental Results

## A.1 Identifying FairQueue as the optimal combination

Table 2: Analysis of **all possible different combinations** for Prompt Queuing (PQ) and tSA Attention Amplification (AA). We summarize our findings from main paper for the tSA "Smiling". Note that $\alpha(S)$ notates AA for tSA tokens, ITI-GEN prompt $P=[T;S]$, and results in bold and italics are the best and second best. Notice that C6:FairQueue (PQ+AA) provides the best combination: it achieves both outstanding sample quality (C6: TA=0.674 & FID=80.02 similar to C1: TA=0.681 & FID=76.9 with the best quality but poor fairness) and fairness (C6: FD=0.069 similar to C4: FD=0.05 with the best fairness but poor quality).

| | Prompt Queuing (PQ) | Attention Amplification (AA) | Stage 1 (Prompt) | Stage 2 (Prompt) | FD($\downarrow$) | TA($\uparrow$) | FID($\downarrow$) | DS($\downarrow$) | remarks |
|---|---|---|---|---|---|---|---|---|---|
| **C1: no PQ, no AA** for Base Prompt | No | No | $T$ | $T$ | 0.211 | **0.681** | **76.9** | - | |
| **C2: no PQ, no AA** for ITI-Gen | No | No | $[T;S]$ | $[T;S]$ | 0.124 | 0.605 | 88.63 | 0.557 | |
| **C3: AA only** for Base Prompt | No | Yes | $T$ | $T$ | N.A | N.A | N.A | N.A | Unimplementable combination due to the absence of tSA tokens for AA |
| **C4: AA Only** for ITI-Gen | No | Yes | $[T;S]$ | $[T;\alpha(S)]$ | **0.05** | 0.610 | 89.41 | 0.550 | |
| **C5: PQ Only** | Yes | No | $T$ | $[T;S]$ | 0.145 | *0.674* | 80.15 | **0.240** | |
| **C6: PQ + AA** (Our proposed: FairQueue) | Yes | Yes | $T$ | $[T;\alpha(S)]$ | *0.069* | 0.674 | *80.02* | *0.284* | Both PQ and AA are present *i.e.,* FairQueue |

In this section, we discuss in more detail how we identified FairQueue – with its two mechanisms: Attention Amplification and Prompt Queuing – as the best-performing solution. Specifically, we summarize our findings, discussed throughout the paper, when exhaustively considering all possible combinations. Our results are as follows:

- **C1: Base Prompt $T$ Only** (no AA no PQ): It lacks tSA-specific knowledge and results in poor fairness. Additionally, without tSA tokens $S$, **AA is not applicable for C3.**

- **C2: ITI-Gen prompt $P$ Only,** in Tab. 1 (no AA no PQ). Our analysis in Sec 3.2 shows it has poor quality due to distortion in global structure during sample generation. Without PQ, the issue of distorted global structure persists for some tSAs

- **C4: Attention Amplification (AA) Only**, in Fig. 5 when PQ transition point=0. It results in poor quality since only ITI-Gen is used. We remark that utilizing only AA for ITI-Gen may deceptively improve fairness, but the generated samples have poor quality e.g., Smiling cartoons. The reason is (similar to C2): without PQ, the issue of distorted global structure persists for some tSAs.

- **C5: Prompt Queuing (PQ) Only,** in Fig. 5 when $c = 0$. By replacing the distorted ITI-Gen prompt with the Base prompt in Stage 1, PQ leads to improved quality, but without AA, the fairness remains poor given reduced exposure to tSA tokens in the denoising process.

- **C6: FairQueue (PQ+AA)**, in Tab 1 Our proposed solution with optimal quality and fairness. Specifically, it combines the effects of Prompt Queuing– enabling the global structure to be properly formed resulting in good quality samples, and Attention Amplification–enhancing the tSA-specific expression for better fairness.

Overall, our results in Tab. 2 reveal that FairQueue (C6) is the superior combination balancing between fairness and quality. Specifically, Prompt Queuing is necessary whereby utilizing either only ITI-GEN (C2) or only Base Prompt (C1) results in quality and fairness degradation, respectively. Furthermore, our results show that both PQ and AA are necessary to obtain high-quality samples with good fairness performance, as without PQ (C4) sample quality is poor, and without AA (C5) fairness performance is degraded.

## A.2 Cross-attention analysis

Sec. 3.2 analyzes the effect of inclusive tokens $S^k$ by comparing the accumulated cross-attention maps of individual tokens between HP and ITI-GEN . It is observed that distorted tokens learned by ITI-GEN negatively affect the development of global structure in the early steps of denoising. The destructive effect arises with abnormally high activity of non-tSA tokens (e.g., ``of''), and the tSA-tokens attend to unrelated regions with scattered attention. A quantitative analysis is performed over 500 sample generations for different tSAs to affirm the observations. The below details how token-specific accumulated cross-attention maps are obtained from the SD pipeline, discusses interaction among tokens, and presents additional representative results for tSAs `Smiling`, `High Cheekbones`, `Gray Hair`, and `Chubby`.

**Details of visualizing token-specific accumulated cross-attention map.** Cross-attention is often used to contextualize prompt embeddings with latent representations per sample generation step. Following DAAM [41], coordinate-aware attention scores $M[S_i]$ are extracted from the latent diffusion network (i.e., U-Net) for the token $S_i$ at the layers where cross-attentions take place. These token-specific attention scores, each with the same spatial dimensions as the latent representation, are upscaled bicubically to the image size ($512 \times 512$ in this case) to reveal where attention is paid per token and accumulated within the assigned step(s). The resulting 2D matrix is visualized in Fig. 3 and Fig. 4 and referred to as an "accumulated cross-attention map".

**Interaction among tokens.** We remark that the cross-attention map of a given token is dependent on the others in the prompt. There are two channels where the effect of tokens may interact: 1) via latent representation, as it is a function of input tokens and serves as the query in the cross-attention (see Sec.2); 2) softmax operation, as a component in the attention pipeline, softmax is taken across all tokens when processing attention scores. These two effects become increasingly apparent as we move through different cross-attention layers of the U-Net and perform more denoising steps.

**HP vs. ITI-GEN : qualitative analysis.** To investigate potential abnormality of ITI-GEN embeddings, images of different tSA are generated conditioning on HP ($F$) and ITI-GEN ($P$) respectively. The cross-attention map is employed as a tool to explore the cause of degraded generations. In pursuit of a fair token-to-token comparison, for some tSAs the original HPs ("HP1", see Tab. 6) are extended to align with $P$ in the number of tSA tokens ("HP2"). Nonetheless, as one can find in the samples in Fig.7 to 14, the extension does not change the behavior of HPs significantly.

Fig.7 to 17 give an overview of cross-attention maps during the denoising process. One may find that the tSA tokens in the HP(s) tend to concentrate on the region(s) semantically associated with the tSA, e.g, mouth for tSA `Smiling`, cheek for tSA `High Cheekbones`, and hair for tSA `Gray Hair`. On the other hand, ITI-GEN tSA tokens' activity tends to be less focused and attends broadly. With more steps than Fig. 3, it is clearer that the global structure of the images is synthesized in the early steps, which motivates the prompt switching experiments.

**Prompt switching experiments and quantitative analysis.** To further investigate HP and ITI-GEN prompts' behaviors in the early steps, the prompt switching experiments (i.e., I2H and H2I) are proposed in Sec. 3.2. Fig.18 to 21 present representative outcomes of the experiments. One can find that the destructive effect caused by ITI-GEN prompts only occurs at the early steps, i.e., Stage 1 in the figures.

In addition, the activation patterns are more clear in the accumulated cross-attention maps. The non-tSA tokens in ITI-GEN prompts are in general more active, and the tSA tokens tend to attend more broadly, which may explain the drastic semantic deviations from HPs in Fig.7 to 17. The latter observation is particularly evident for tSA `Smiling`, a highly localized facial expression, which is supported by the histogram of central moments in Fig.1. The other tSAs, though may not be directly associated with a specific facial feature, share the same trend, as manifested statistically by the histograms in Fig.22 and Fig.23.

**Utilizing Base Prompt ($T$) in FairQueue .** In Prompt Queuing the use of $T$, in place of the HP, is similarly grounded on the I2H/H2I analysis, as both $T$ and HP are natural language prompts – free of learned tokens. This can be seen in the embedding analysis in Supp B.6 where the HP and $T$ are seen to be close to one another. As a result, the sample generated by $T$ is expected to be of similar quality as the HP.

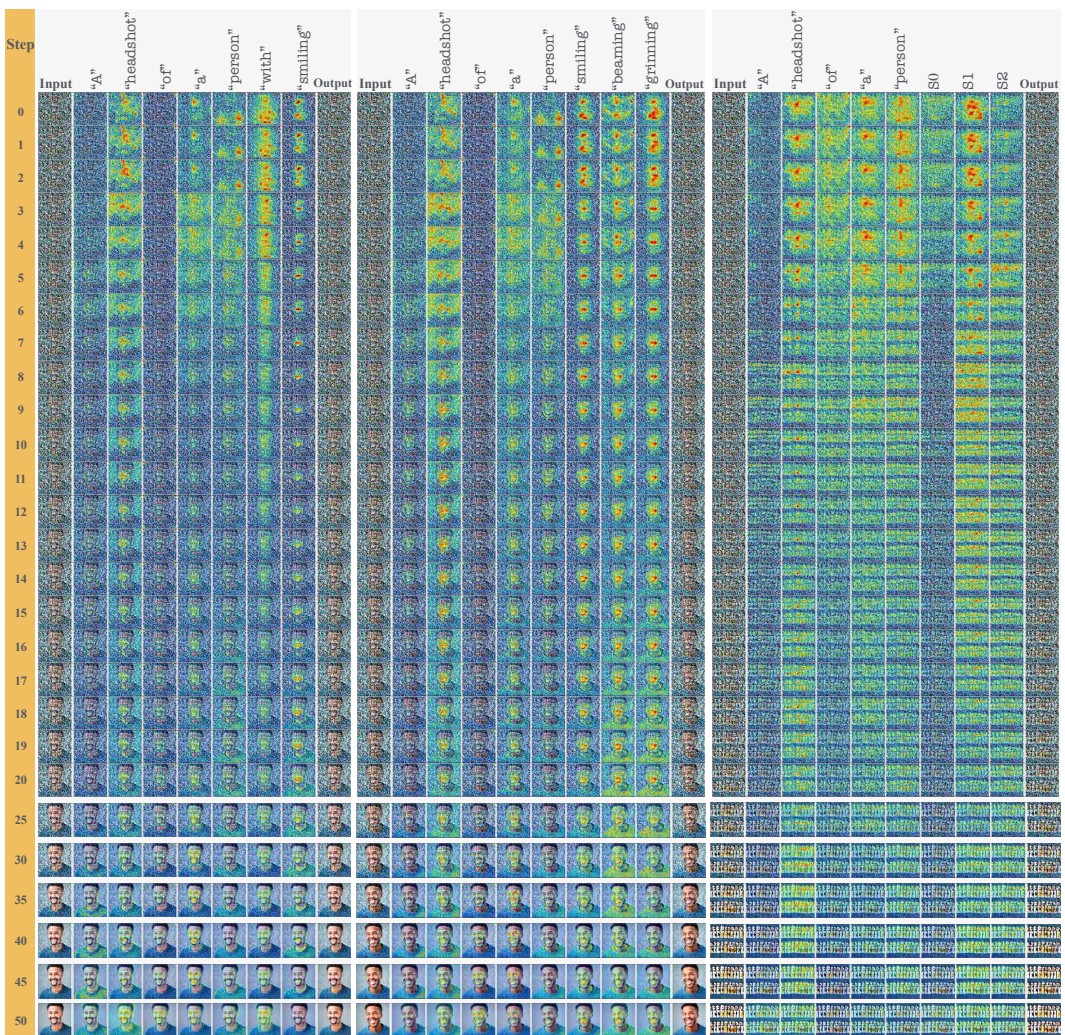

Figure 7: **Cross-attention maps during the denoising process with HP1 (left), HP2 (middle, equal #tSA tokens to ITI-GEN ), and ITI-GEN (right) prompts.** tSA=Smiling.

In Fig. 24, we provide further visualizations of $T$'s effectiveness in generating the global structure in early denoising steps. Specifically, we compare the cross-attention maps of FairQueue with ITI-Gen during sample generation, together with quantitative analysis. Results in col 2 vs 3 illustrate $T$'s effectiveness in synthesizing the global structure in stage 1, and non-abnormal attention (in Fig. 25), resulting in effective global synthesis than ITI-Gen and better sample quality.

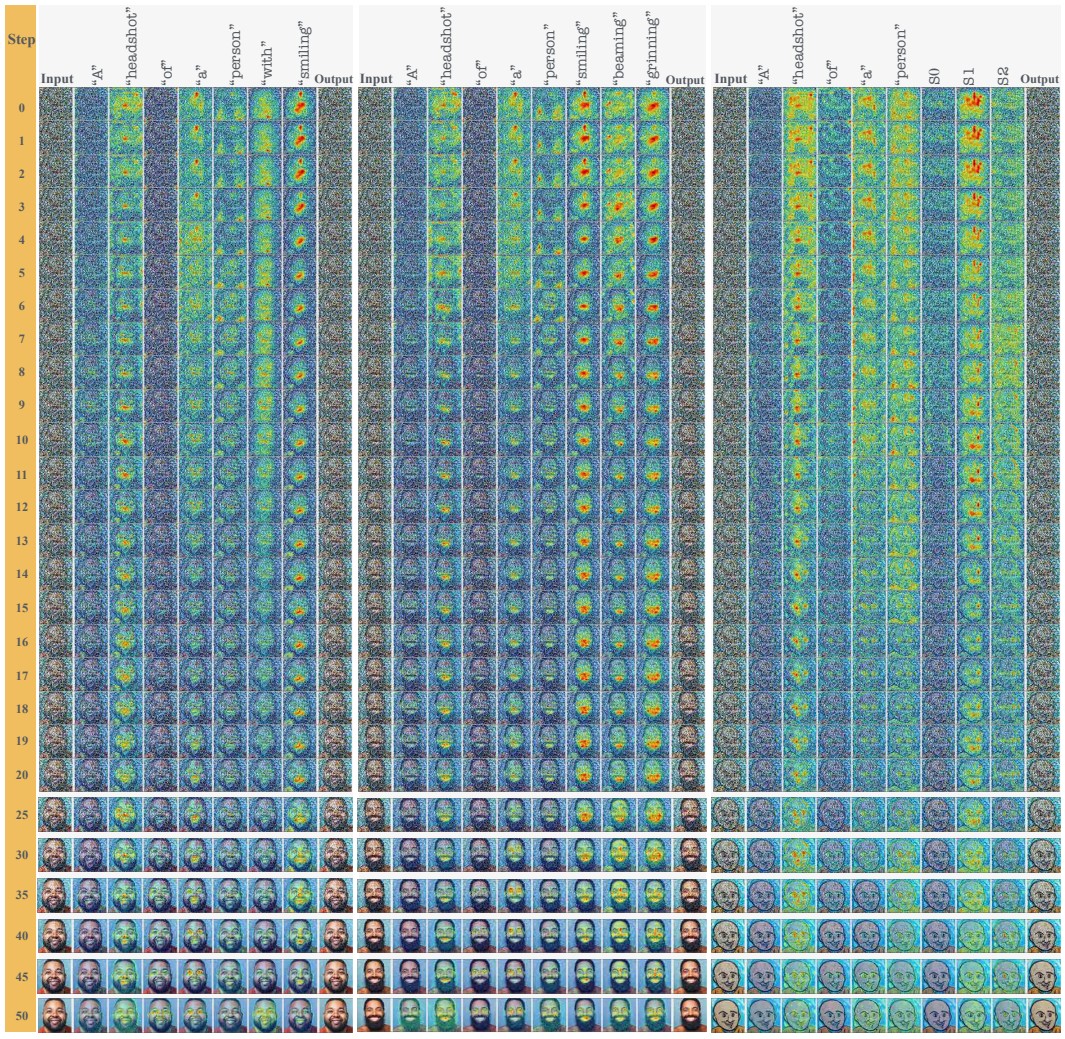

Figure 8: **Cross-attention maps during the denoising process with HP1 (left), HP2 (middle, equal #tSA tokens to ITI-GEN ), and ITI-GEN (right) prompts.** tSA=Smiling.

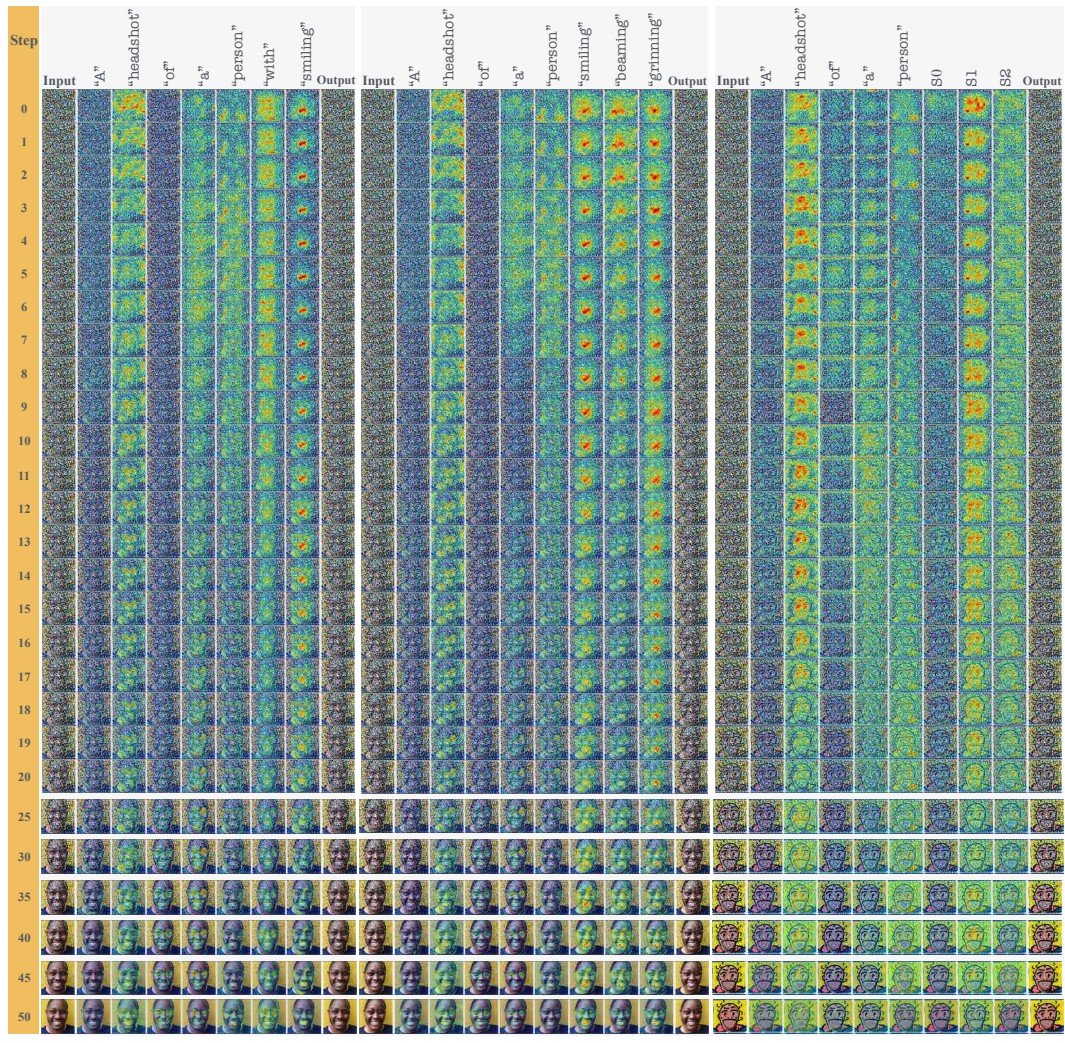

Figure 9: **Cross-attention maps during the denoising process with HP1 (left), HP2 (middle, equal #tSA tokens to ITI-GEN ), and ITI-GEN (right) prompts.** tSA=Smiling.

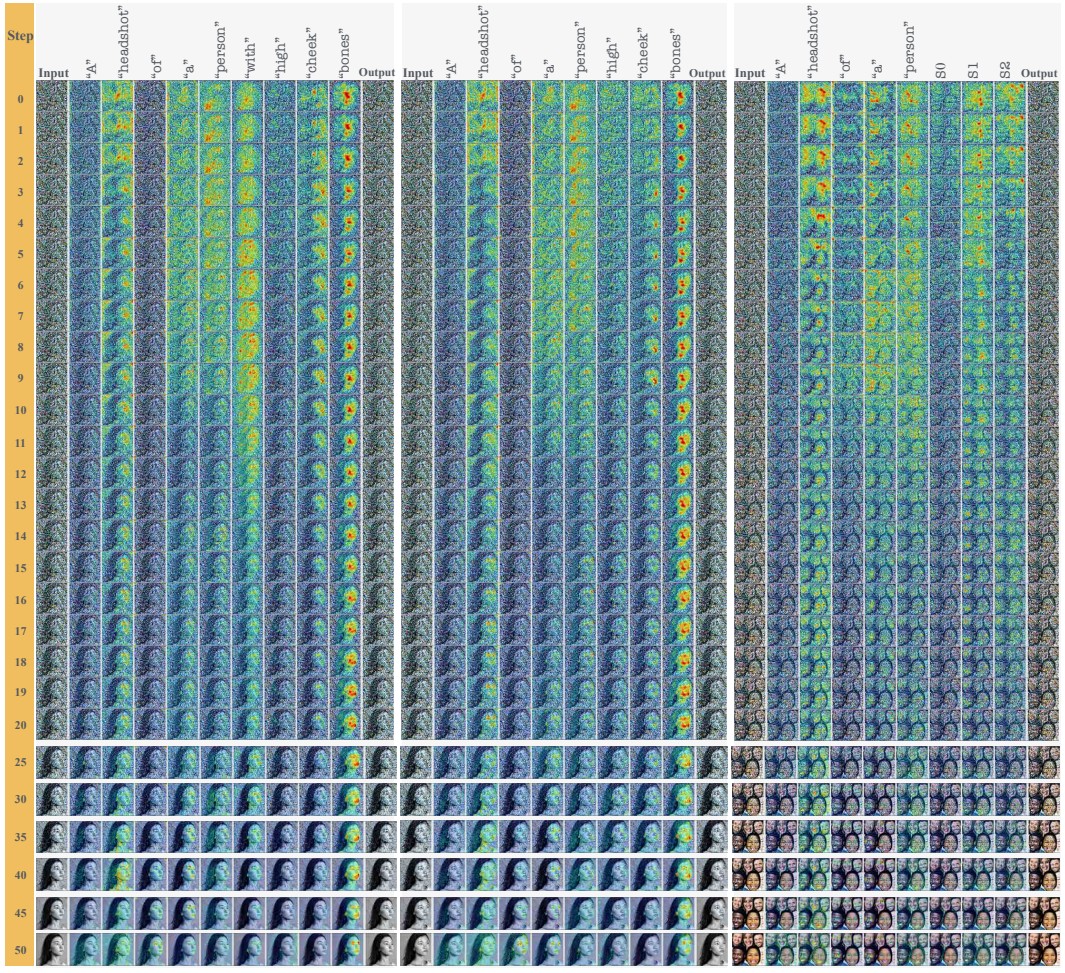

Figure 10: **Cross-attention maps during the denoising process with HP1 (left), HP2 (middle, equal #tSA tokens to ITI-GEN ), and ITI-GEN (right) prompts.** tSA=High Cheekbones.

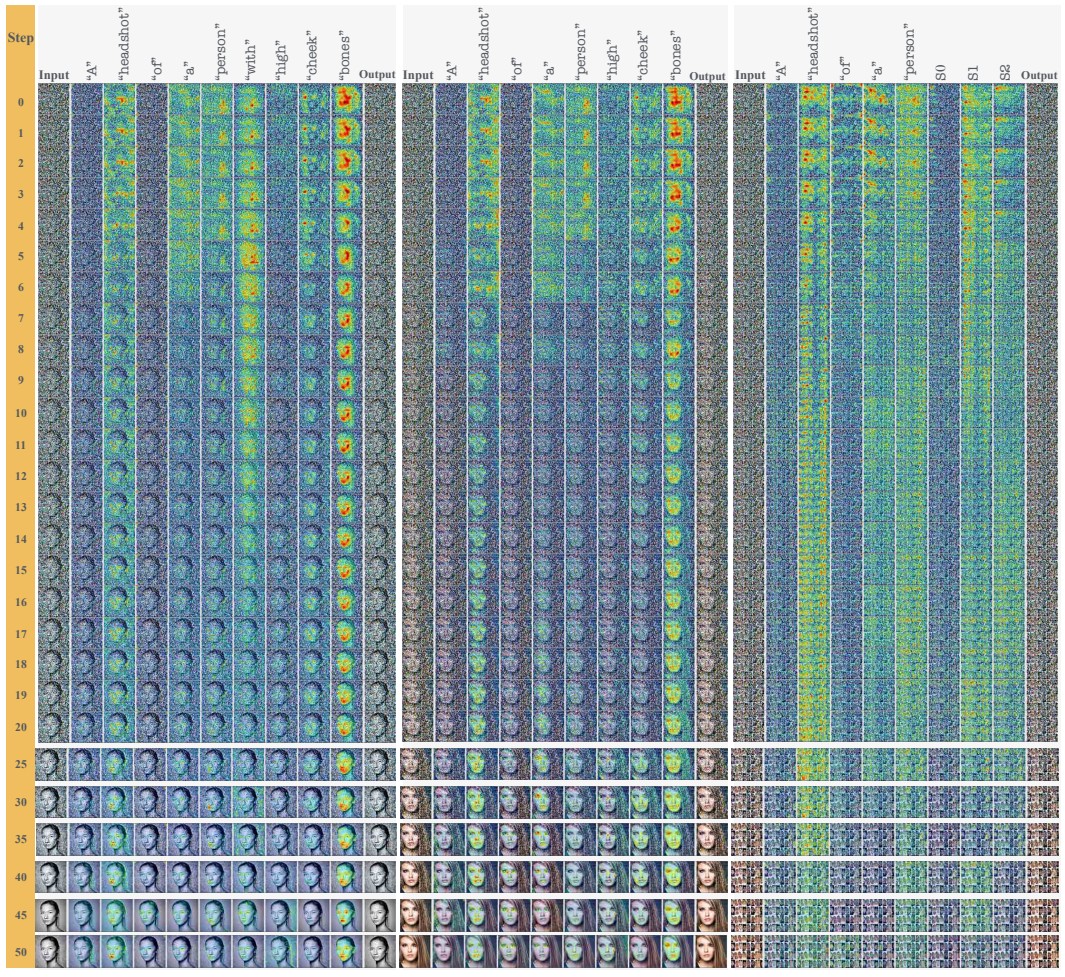

Figure 11: **Cross-attention maps during the denoising process with HP1 (left), HP2 (middle, equal #tSA tokens to ITI-GEN ), and ITI-GEN (right) prompts.** tSA=High Cheekbones.

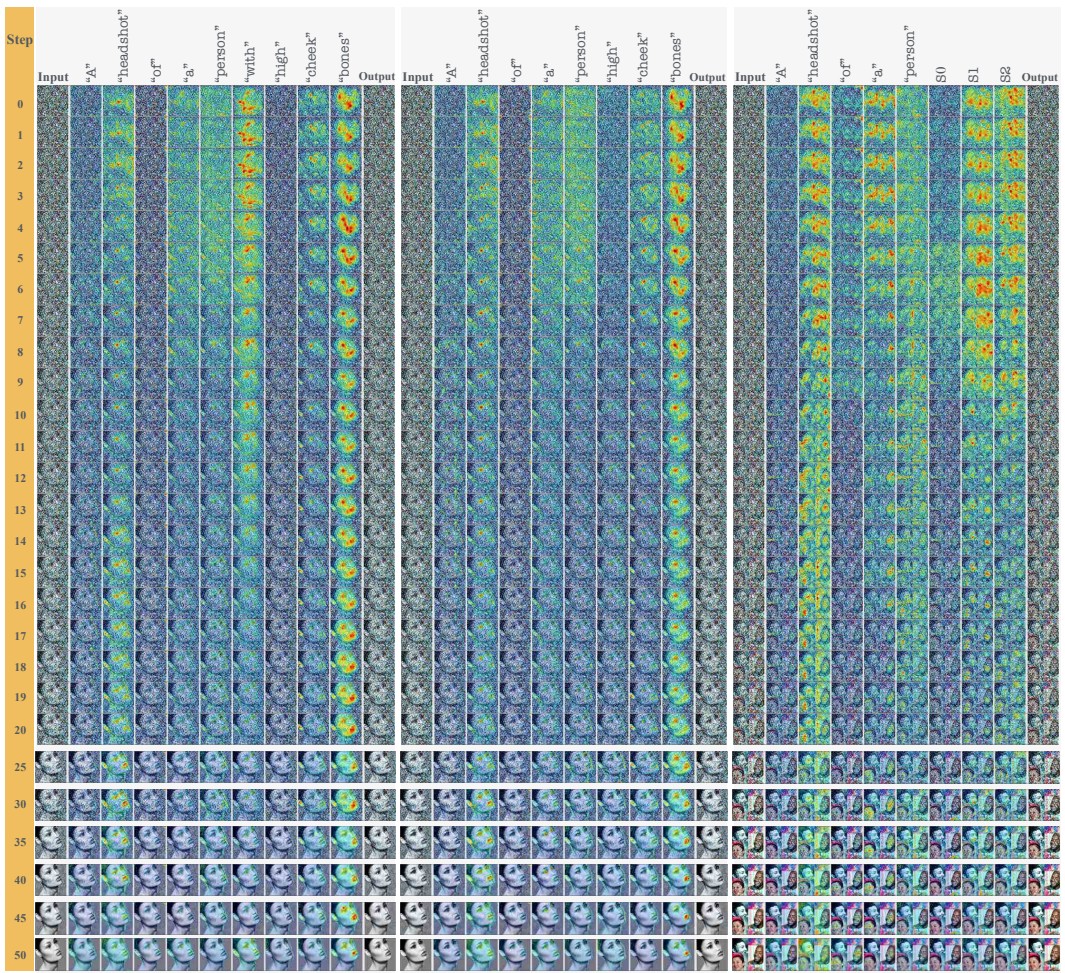

Figure 12: **Cross-attention maps during the denoising process with HP1 (left), HP2 (middle, equal #tSA tokens to ITI-GEN ), and ITI-GEN (right) prompts.** tSA=High Cheekbones.

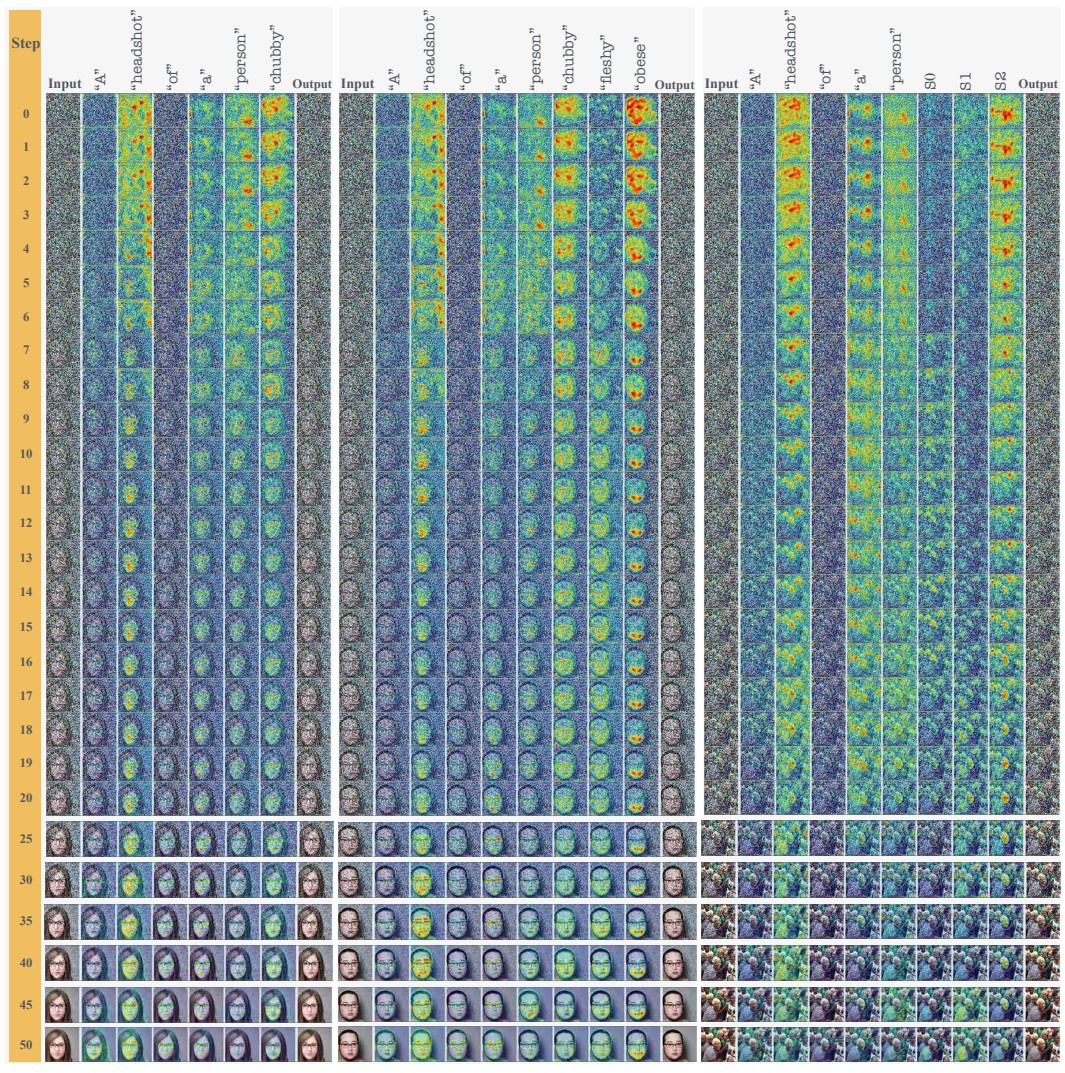

Figure 13: **Cross-attention maps during the denoising process with HP1 (left), HP2 (middle, equal #tSA tokens to ITI-GEN), and ITI-GEN (right) prompts.** tSA=Chubby.

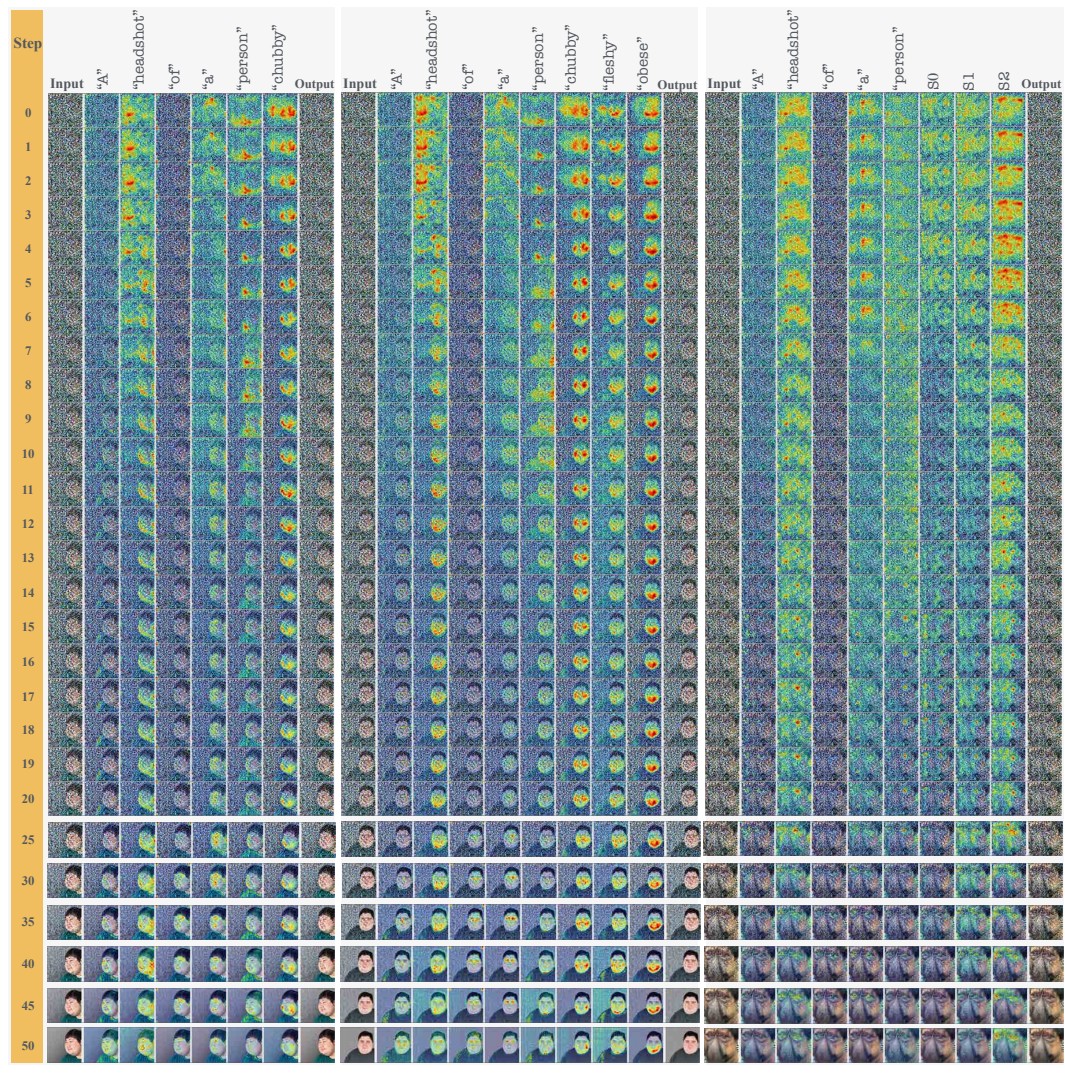

Figure 14: **Cross-attention maps during the denoising process with HP1 (left), HP2 (middle, equal #tSA tokens to ITI-GEN ), and ITI-GEN (right) prompts.** tSA=Chubby.

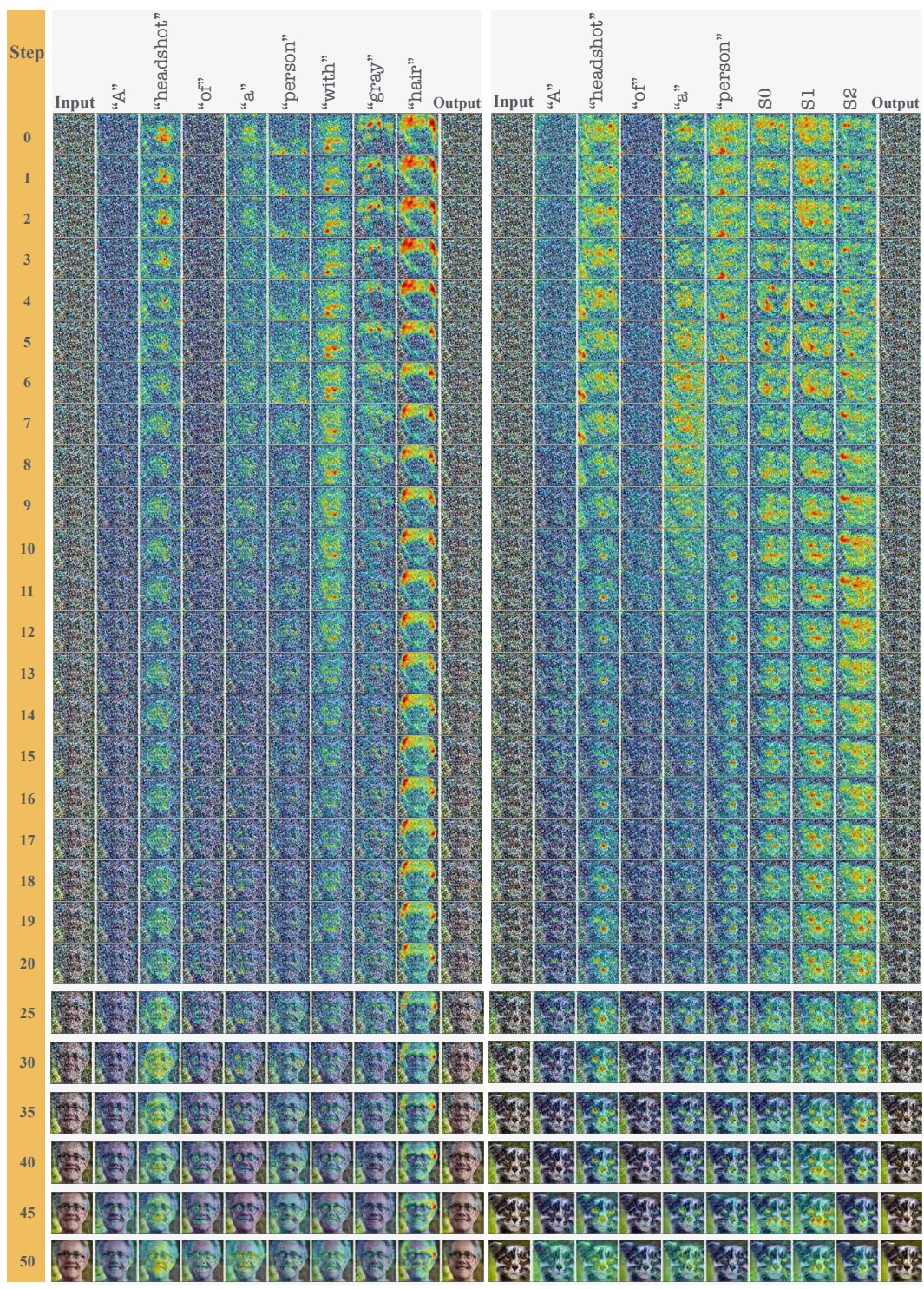

Figure 15: **Cross-attention maps during the denoising process with HP (left) and ITI-GEN (right) prompts.** tSA=Gray Hair.

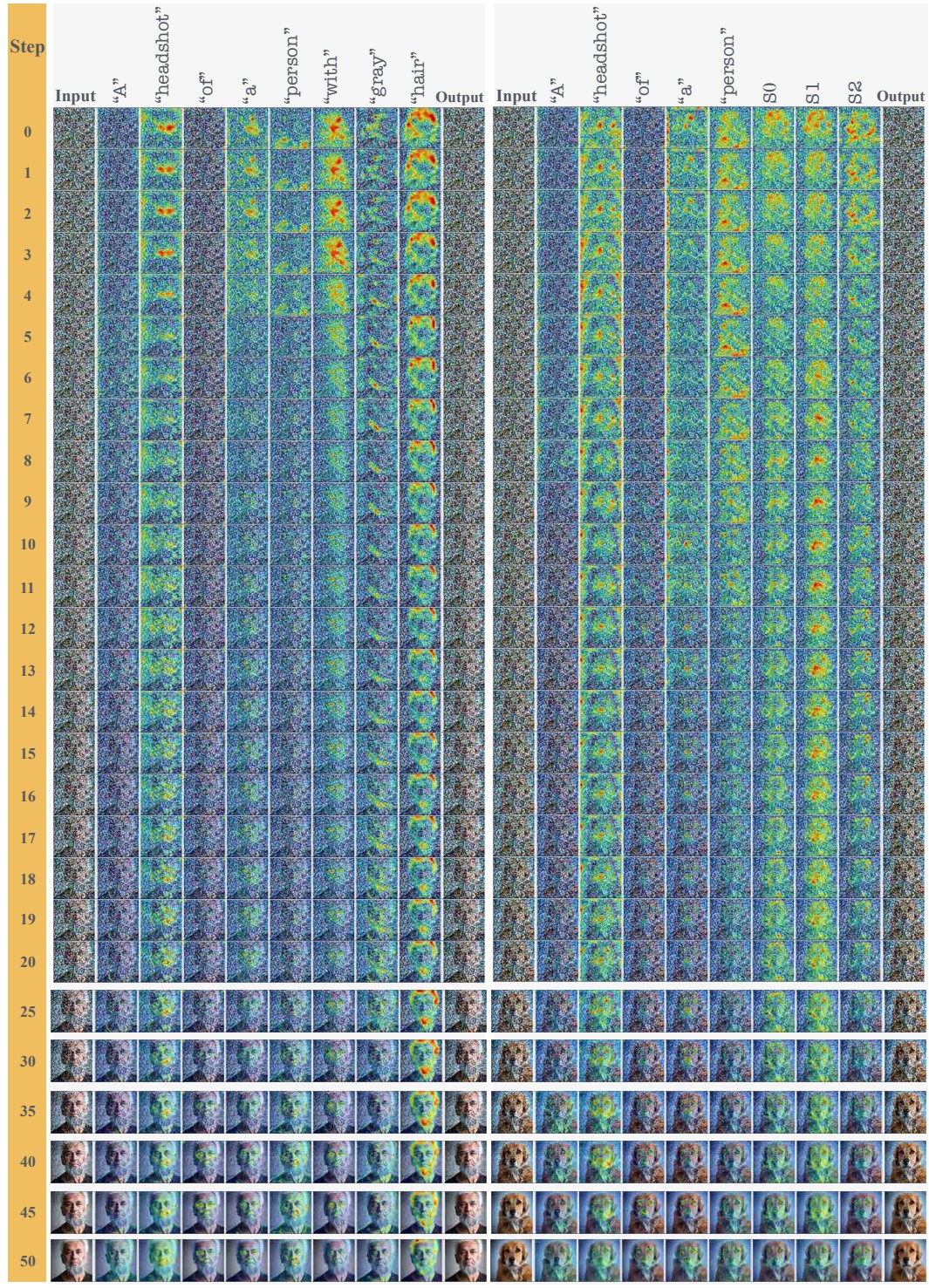

Figure 16: **Cross-attention maps during the denoising process with HP (left) and ITI-GEN (right) prompts.** tSA=Gray Hair.

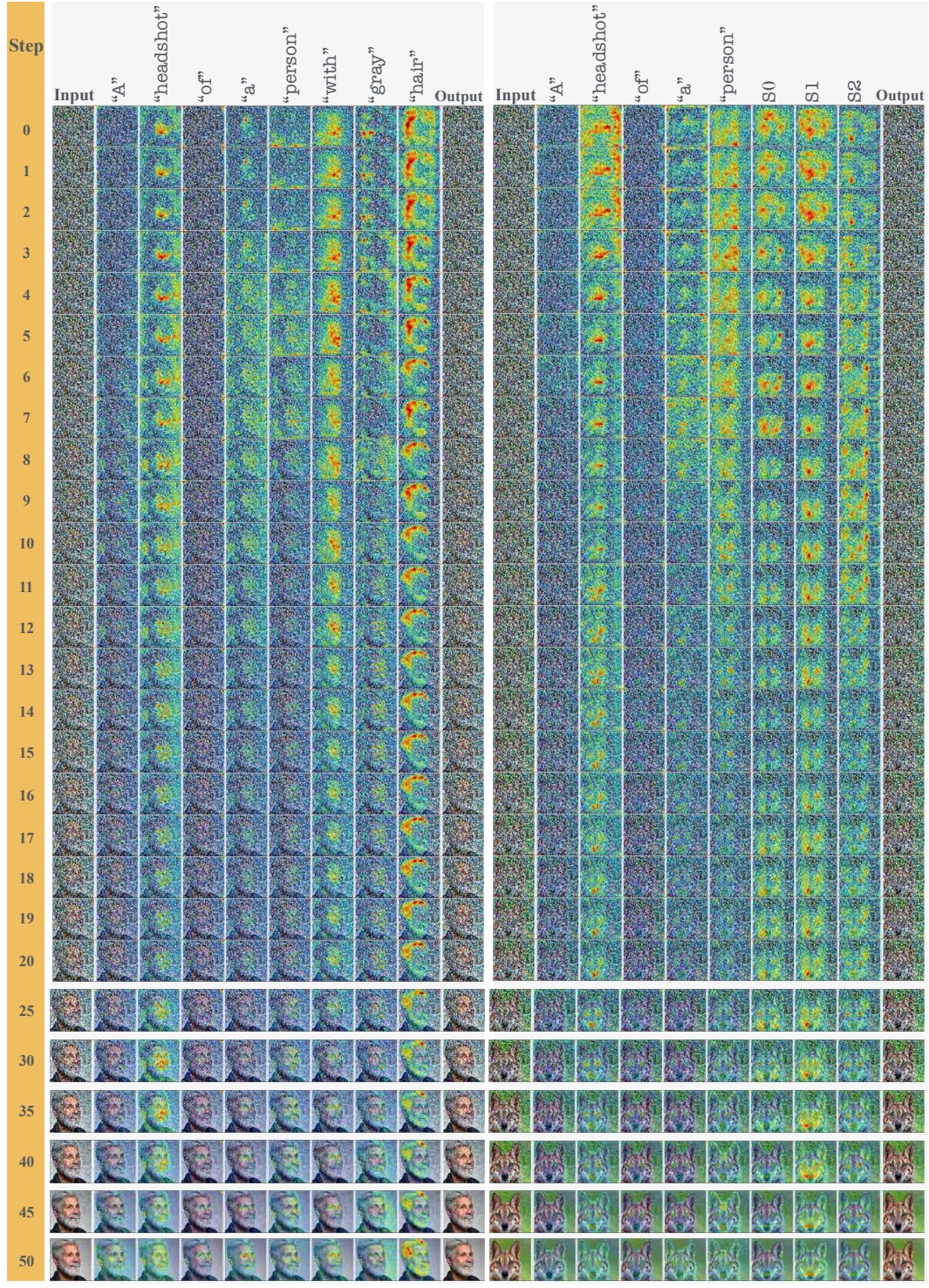

Figure 17: **Cross-attention maps during the denoising process with HP (left) and ITI-GEN (right) prompts.** tSA=Gray Hair.

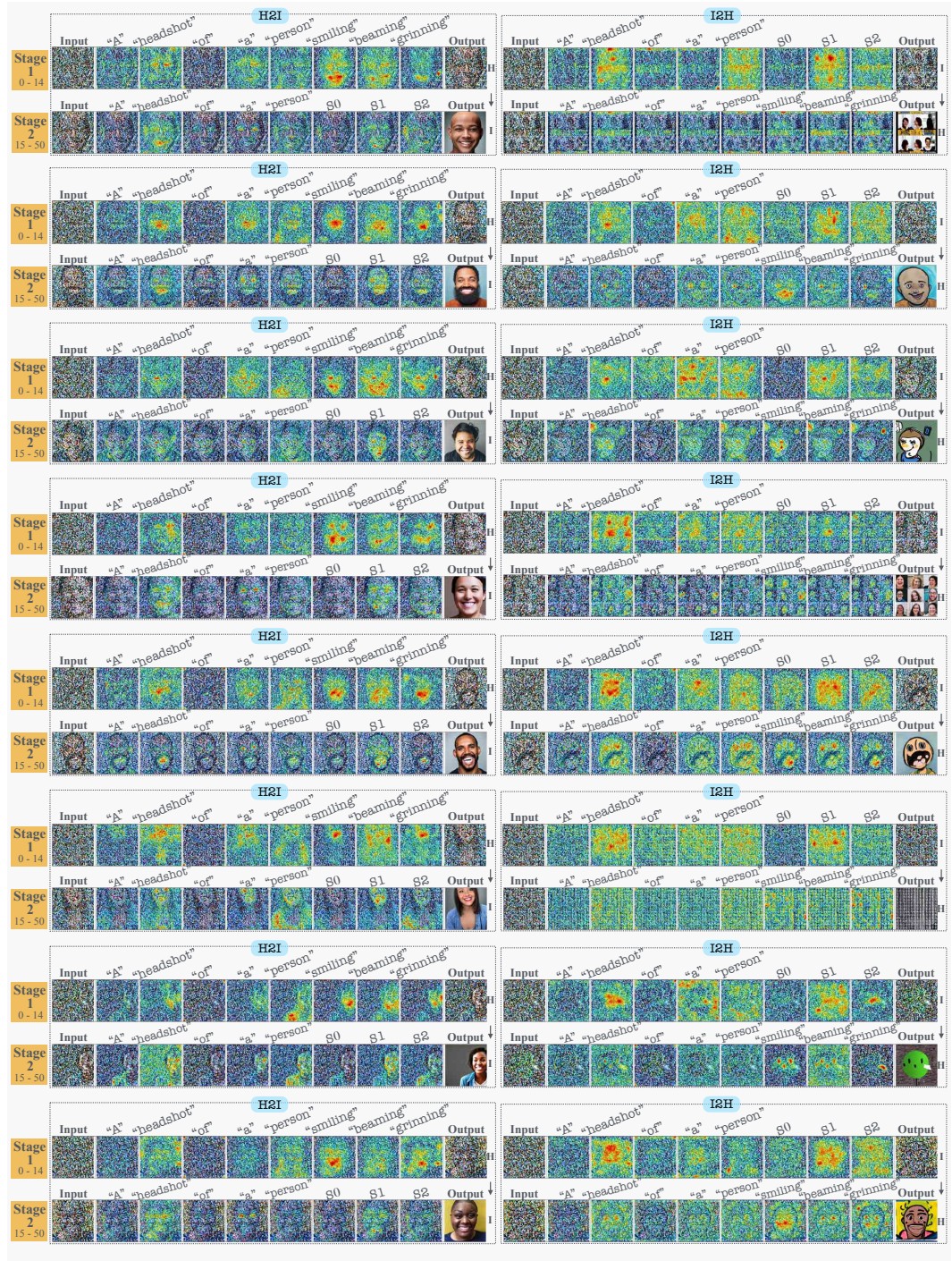

Figure 18: **Accumulated cross-attention maps for the denoising process in our proposed prompt switching experiments I2H and H2I: tSA =** `Smiling`**.**

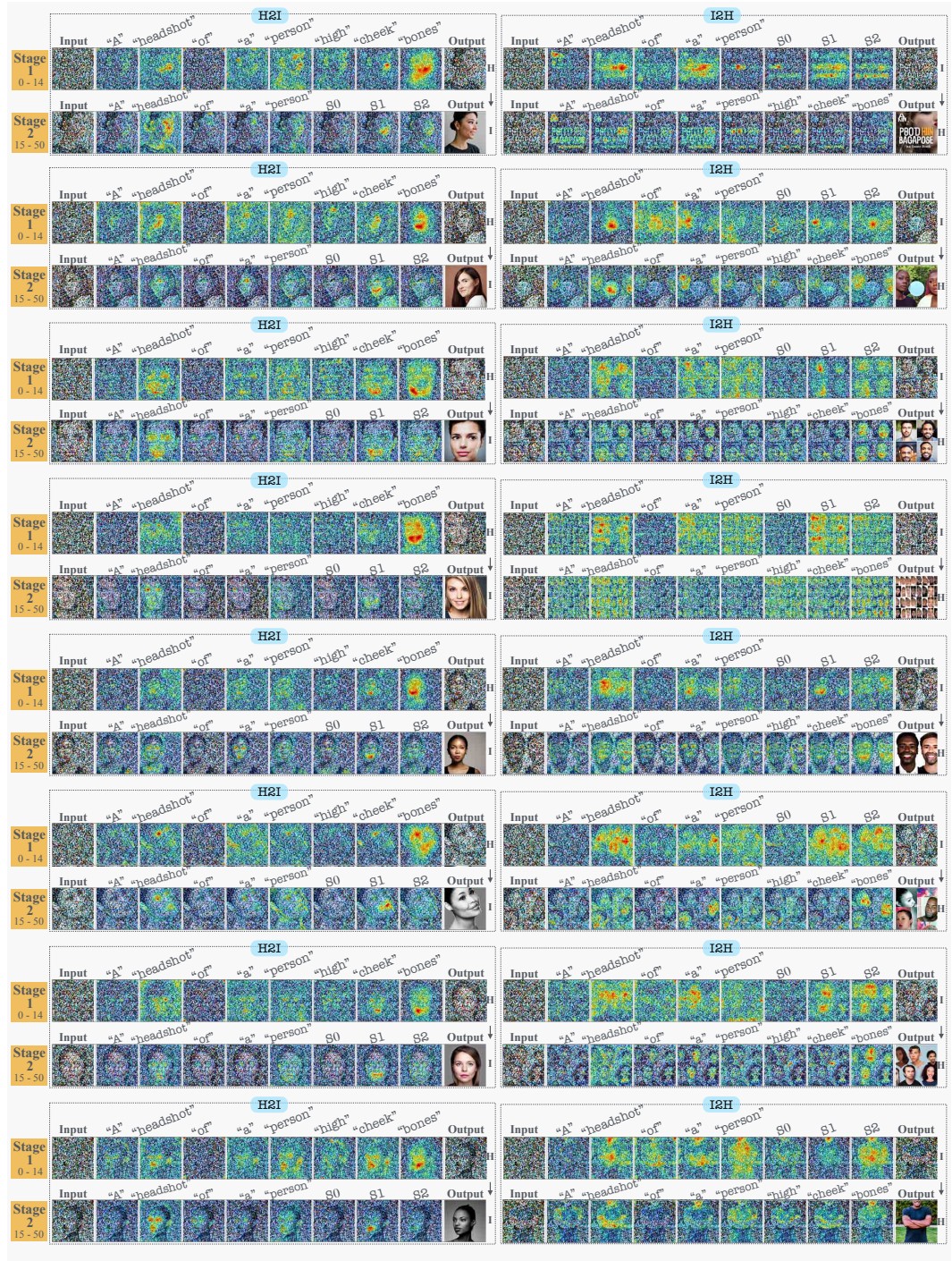

Figure 19: **Accumulated cross-attention maps for the denoising process in our proposed prompt switching experiments I2H and H2I: tSA =** `High Cheekbones`**.**

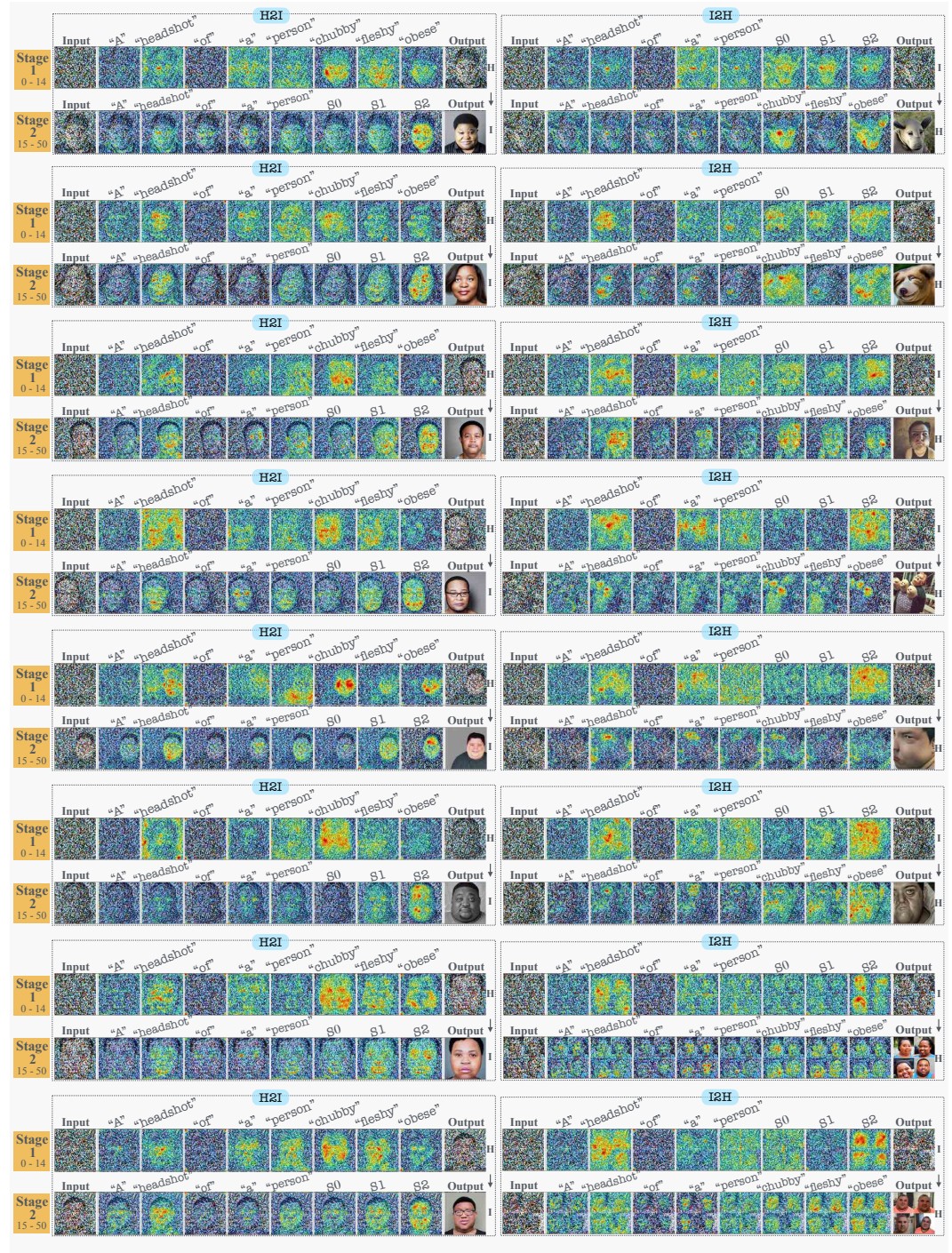

Figure 20: **Accumulated cross-attention maps for the denoising process in our proposed prompt switching experiments I2H and H2I: tSA =** Chubby**.**

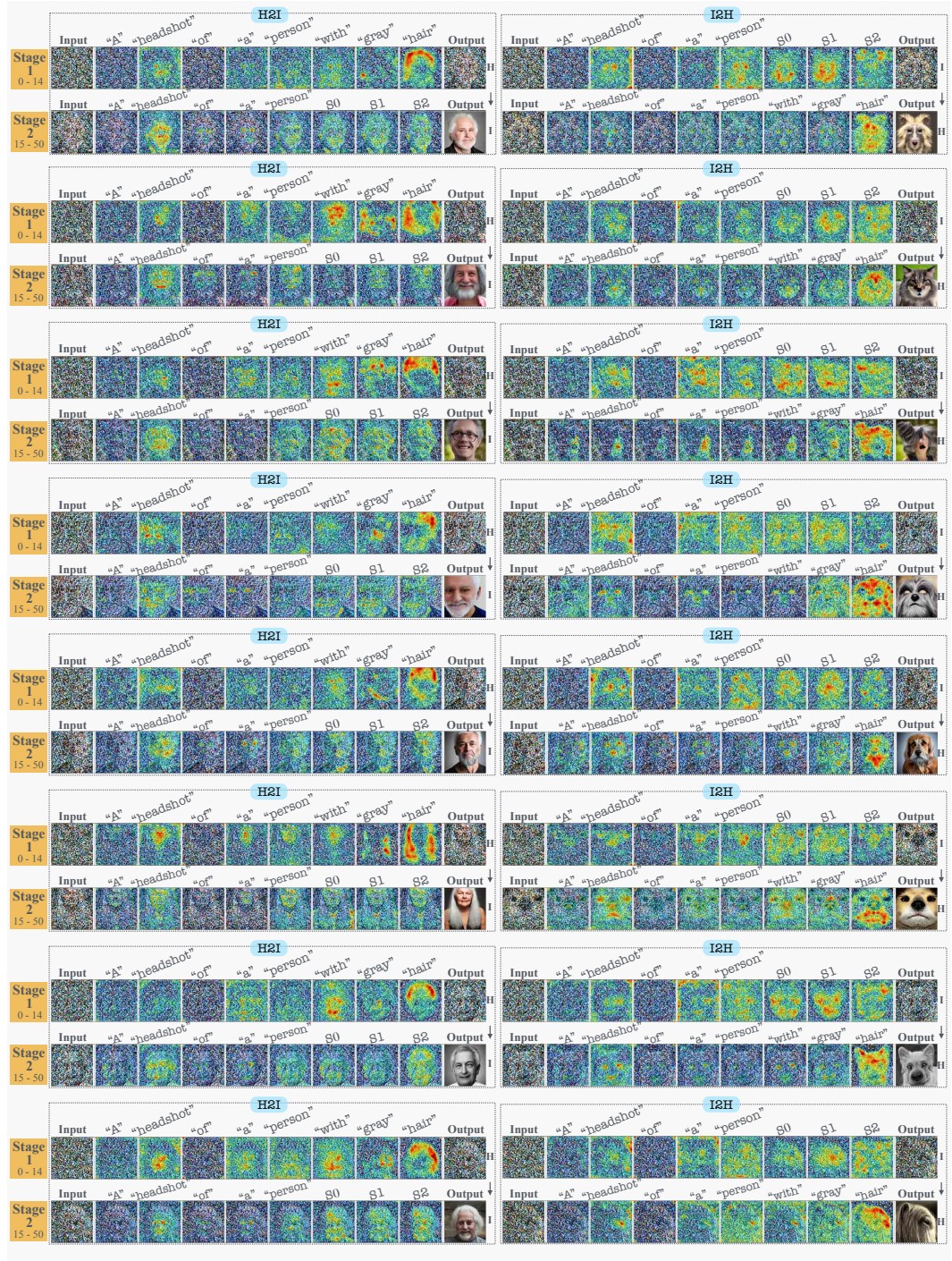

Figure 21: **Accumulated cross-attention maps for the denoising process in our proposed prompt switching experiments I2H and H2I: tSA =** `Gray Hair`**.**

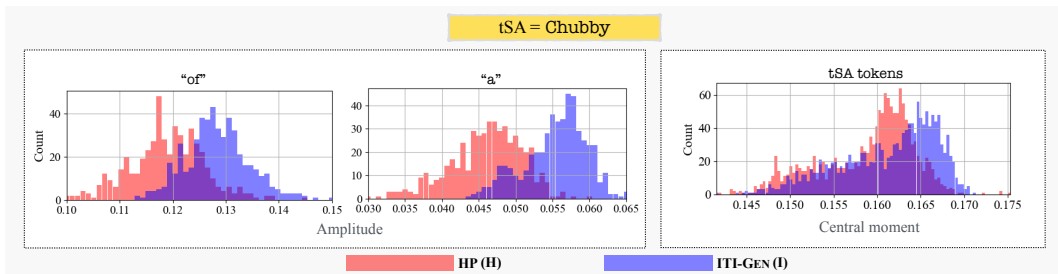

Figure 22: **Histograms for cross-attention analysis in prompt switching experiments I2H and H2I: tSA =** Chubby**.**

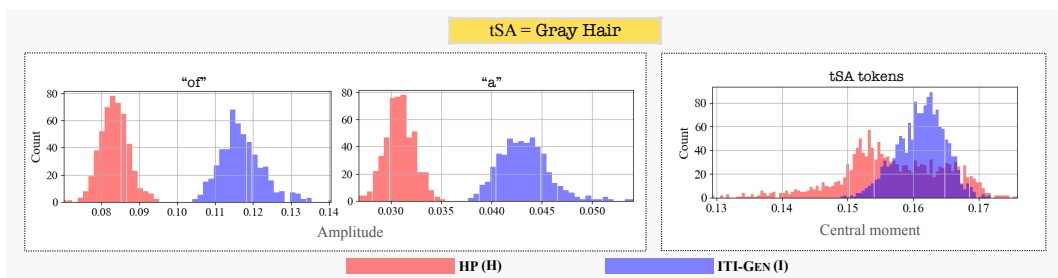

Figure 23: **Histograms for cross-attention analysis in prompt switching experiments I2H and H2I: tSA =** Gray Hair**.**

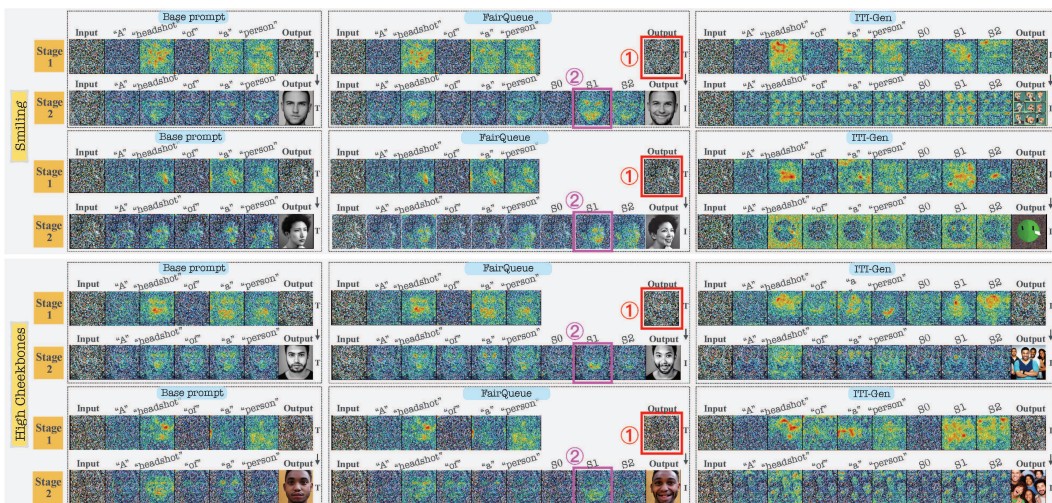

Figure 24: Accumulated cross-attention maps for Base prompt, FairQueuing and ITI-Gen, tSA=Smiling, High Cheekbones. Same setup as Sec 3.2 of the main manuscript (e.g., Fig.4). The mid column presents FairQueue, with Base prompt (T) in Stage 1 and ITI-Gen (I) in Stage 2; the left column is only based on Base prompt and the right is only based on ITI-Gen. Note Base prompt behaves similarly to the HP in forming good global structures in the first stage (annotated by red frames); in the second stage, the tSA token of ITI-Gen can attend to tSA-related regions (e.g., eyes and mouth for Smiling, or the lower half of the face and cheeks for High Cheekbones, see magenta frames) and enhance associated facial features.

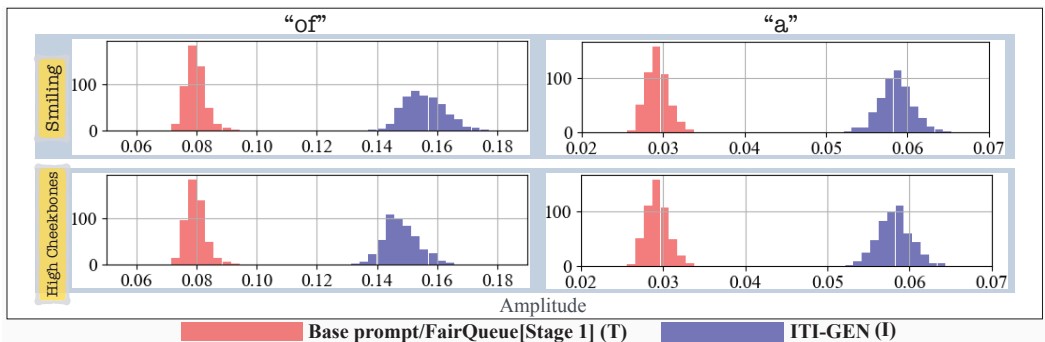

Figure 25: Histograms for non-tSA tokens in the first stages, tSA=Smiling and High Cheekbones for Fig. 24.

### A.3 More on Ablation Studies

### A.3.1 Analyzing the Effects of Attention Amplification

In this section, we provide more illustrations for the ablation study. Our results in Fig. 26, illustrate the effect of attention application with different scaling factors ($c$). Notice that at $c = 0$ the tSA expression may still be lacking but increasing $c$ results in emphasized tSA expression.

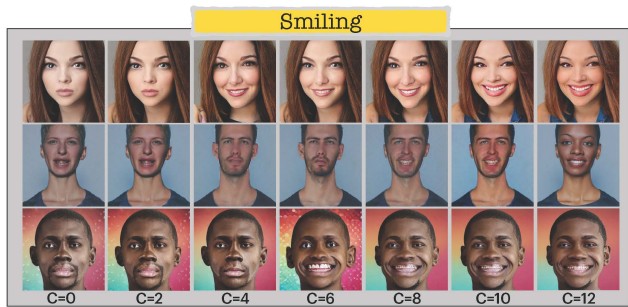

Figure 26: Illustration of FairQueue samples utilizing different attention amplification scaling factors ($c$), tSA=Smiling. In each row, we utilize the same seed and prompt queuing transition point.

### A.3.2 More Illustrations for Training Once-fo-All Tokens

In Fig. 27, we provide more illustrations for the analysis on Revisiting Training Once-for-All Token.

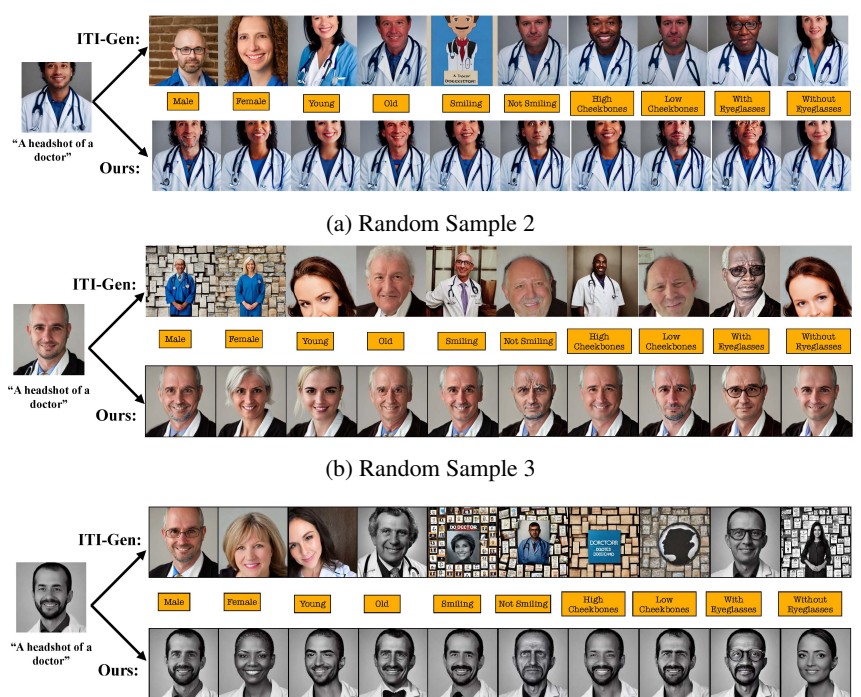

Figure 27: **Illustration of samples generated by ITI-GEN and FairQueue with a new Base Prompt $T' = E_t\{$"A headshot of a doctor"$\}$ via pre-pending.** Notice that FairQueue improved sample quality and ability to preserve the original sample's semantics while mainly adapting only the tSA.

### A.3.3 Human-recognized Assessment Comparing ITI-Gen and FairQueue Quality

In this section, we carried out a human user study to evaluate the sample quality and fairness of the generated sample from FairQueue against ITI-GEN . Specifically, we utilize the same seed to generate 100 sample pairs with ITI-Gen and FairQueue for { Smiling, High Cheekbones, Gender, and Young }. Then, utilizing Amazon Mechanical Turk we conduct 2 tasks:

- **Quality comparison by A/B testing:** Human labelers select the better quality sample between ITI-Gen and FairQueue (from the same seed). Each sample was given to 3 labelers.
- **Fairness comparison by human-recognized tSA:** labelers identified the tSA class for each sample. The final label was based on the majority of 3 labelers. labelers were also given an "unidentifiable" option if the class could not be determined. Finally, the labels were used to measure FD.

Our results in Tab. 3 reveal that FairQueue generates better quality samples than ITI-Gen (>62.0% preference) and Tab. 4 shows that FairQueue achieves competitive fairness with ITI-Gen. Overall, this aligns with our quantitative results in Tab. 1.

Table 3: A/B testing: Human assessment comparing quality between ITI-Gen vs FairQueue for 200 samples per tSA. Col 2 and 3 indicate the percentage of labelers that prefer the method's sample quality. A larger value is better.

|  | ITI-GEN | FairQueue |
|---|---|---|
| Smiling | 1.3% | 98.7% |
| High Cheekbones | 2.7% | 97.3% |
| Gender | 33.0% | 67.0% |
| Young | 38.0% | 62.0% |

Table 4: Fairness comparison by human-recognized tSA: Human assessment to compare FD for ITI-Gen vs FairQueue for 200 samples per tSA.

|  | ITI-GEN FD($\downarrow$) | FairQueue FD($\downarrow$) |
|---|---|---|
| Smiling | 0.106 | 0.014 |
| High Cheekbones | 0.144 | 0.021 |
| Gender | 0.014 | 0.014 |
| Young | 0.014 | 0.028 |

## A.4 More Illustration

In this section, we provide more samples generated by FairQueue based on the setup in Sec. 5. Recall that here we utilize the base prompt $\boldsymbol{T} = E_T$''`A headshot of a person`'' and consider the tSA$\in$ {`Male`, `Young`, `Smiling`, `Low Cheekbones`, `Pale Skin`, `Eyeglasses`, `Mustache`}. Each sample is then generated based on the same 10 fixed noise inputs.

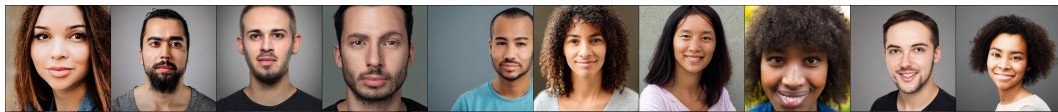

Figure 28: Base-Prompt images $\boldsymbol{T}$ with fixed latent noise input

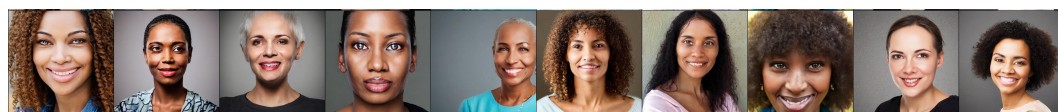

Figure 29: FairQueue with tSA=`Female` with fixed latent noise input

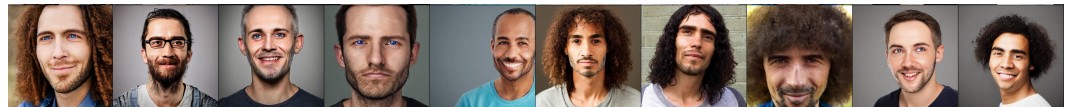

Figure 30: FairQueue with tSA=`Male` with fixed latent noise input

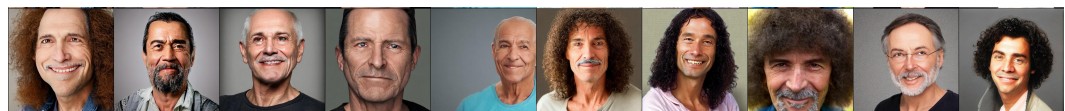

Figure 31: FairQueue with tSA=`Old` with fixed latent noise input

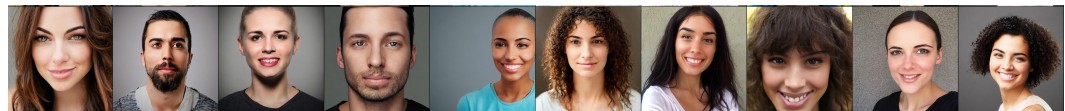

Figure 32: FairQueue with tSA=`Young` with fixed latent noise input

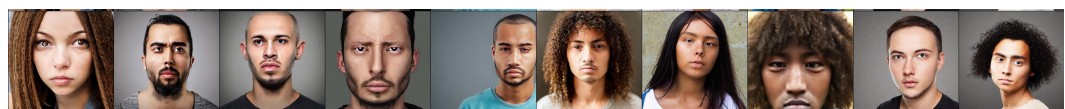

Figure 33: FairQueue with tSA=`not Smiling` with fixed latent noise input

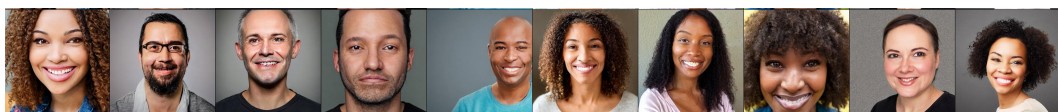

Figure 34: FairQueue with tSA=`Smiling` with fixed latent noise input

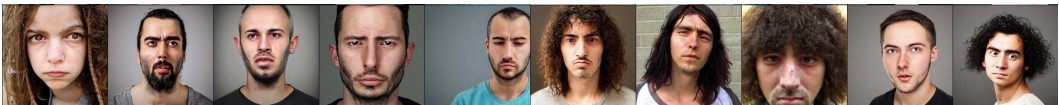

Figure 35: FairQueue with tSA=`Low Cheekbones` with fixed latent noise input

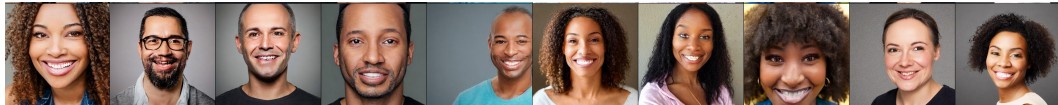

Figure 36: FairQueue with tSA=`High Cheekbones` with fixed latent noise input

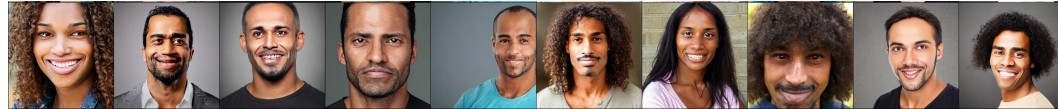

Figure 37: FairQueue with tSA=`not Pale Skin` with fixed latent noise input

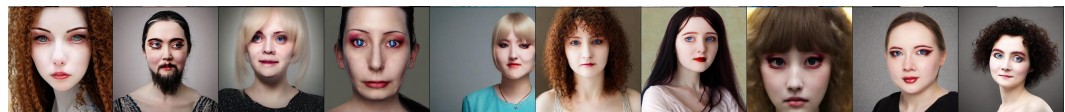

Figure 38: FairQueue with tSA=`Pale Skin` with fixed latent noise input

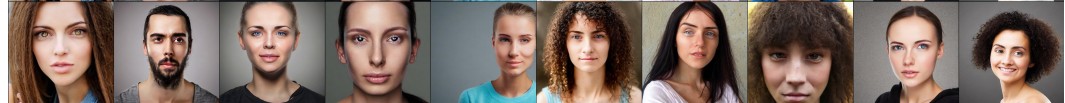

Figure 39: FairQueue with tSA=`no Eyeglasses` with fixed latent noise input

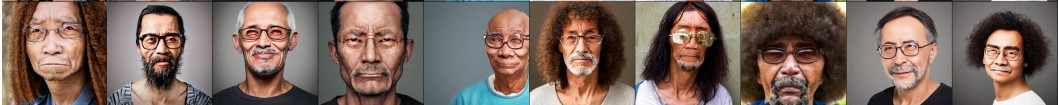

Figure 40: FairQueue with tSA=`with Eyeglasses` with fixed latent noise input

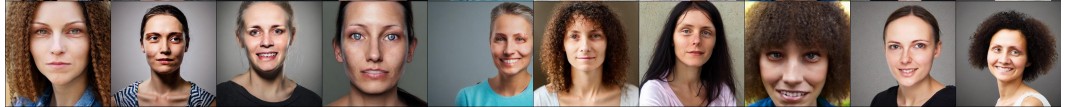

Figure 41: FairQueue with tSA=`no Mustache` with fixed latent noise input

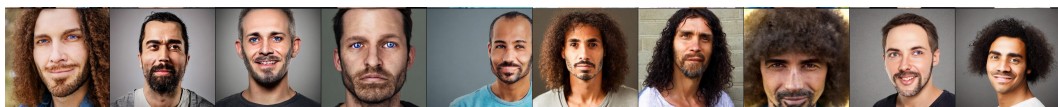

Figure 42: FairQueue with tSA=`with Mustache` with fixed latent noise input

## A.5 Evaluating Minimal Linguistic Ambiguity for tSA

In this section, we provide an experiment to search for tSA with minimal linguistic ambiguity. To do this, we utilize the tSA and their respective HPs in Tab. 6. Then, with these HPs, we generate 500 samples per category of the individual tSA. Finally, we classify the respective samples and evaluate the accuracy of the T2I model in generating samples with the respective categories of the tSA. Our results in Tab. 5 report the accuracy of the T2I generative model in accurately

interpreting the HPs to generate the respective tSAs. Based on our results, we can determine that the tSA$\in \{$Male, Young, Smiling, High Cheekbones$\}$ are with minimal linguistic ambiguity, as their HPs can be easily interpreted by the T2I generator as seen by the high accuracy.

Table 5: Accuracy of the T2I model in generating the tSA with the HP. See Tab. 6 for list on all the HPs.

| tSA | Positive Prompts | Negative Prompts |
|---|---|---|
| Male | 99.8% | 99.2% |
| Young | 97.8% | 98.4% |
| Smiling | 98.2% | 97.8% |
| High Cheekbones | 96.6% | 98.8% |
| Pale Skin | 91.3% | 7.8% |
| Eyeglasses | 97.4% | 2.4% |
| Mustache | 90.4% | 13.6% |
| Gray Hair | 97.2% | 22.4% |
| Chubby | 98.2% | 30.4 % |

# B  Experimental Details

## B.1  Details of Calculating Directional Loss for Prompt Tuning

In this section, we provide more details on the directional loss, $\mathcal{L}_{dir}$ utilized by ITI-GEN . Recall that in ITI-GEN the direction loss takes in two components: 1) direction of Image embeddings, $\Delta \boldsymbol{I}$, and 2) direction of ITI-GEN token embedding, $\Delta \boldsymbol{P}$. For ease of discussion, we utilize a binary tSA, but the same concept can be simply scaled to multi-class tSA.

Specifically, we first measure $\Delta \boldsymbol{I}$ where ITI-GEN first evaluates the mean image embedding for each category of the tSA *i.e.,* $\alpha = \mathbb{E}_k \left[ E_I(x^k) \right]$, where $E_I$ is a CLIP Image encoder [20] and $x^k$ are the reference image for $k$ samples in a given mini-batch. Then, we calculate the directional Image embeddings: $\Delta \boldsymbol{I}_{i,j} = \alpha_i - \alpha_j$ where $i$ and $j$ are the different categories of the selected tSA. Then when considering direction of ITI-GEN token embedding we simply calculate $\Delta \boldsymbol{P} = E_T(\boldsymbol{P}_i) - E_T(\boldsymbol{P}_j)$, where $E_T$ is the CLIP text encoder.

## B.2  Details of the Ambiguities in Text Prompts

Ambiguities arise because of the potentially multiple interpretations of the same utterance. Ambiguity is a well-known concept in Large Language Models (LLMs), and recently there have been some attempts to understand the impact of these ambiguities in the intersection of the LLM and image generation models, a.k.a T2I models. Among different types of ambiguity, three major types can affect the quality of the T2I generation [47]:

- *Syntax:* where there could be different interpretations of the same text. As an example, in input prompt ''the mouse looks at the cat standing on rug'', it is not clear whether the mouse is standing on the rug or the cat.
- *Semantics:* where the words within a text have multiple meanings. As an example, ''a photo of a bat'', it is not clear whether it refers to a nocturnal flying mammal or the flat wooden club used in sports like baseball or cricket (with 'cricket' itself having two different meanings ;)).
- *underspecification* where the used text prompt can not completely describe the required attributes in the image. For example, ''a woman with light hair'' can not specifically determine the category of the hair color.

Within the context of this paper, ambiguity is generally used with the Hard Prompts (HP) where we append the description of a specific tSA to the base prompt. For example, considering the base prompt ''a headshot of a person'', and considering the tSA=Smiling, one can create the HP as ''a headshot of a person smiling'', and ''a headshot of a person not smiling''. In our paper, we empirically realize that for some tSA, creating HP like this will result in a strong baseline for fair T2I generation, observed by our quality and fairness metrics. We call these, tSAs with minimum linguistic ambiguity, and as an example, we empirically found that tSAs like Young, Smiling, Gender fall within this category. Due to their superior performance, HP with these tSAs is used as our pseudo-gold standard in the analysis of the quality degradation issue in ITI-GEN is

Sec. 3. However, other tSAs suffer ambiguity in either finding proper terms to define them or inferior performance in terms of the quality or fairness of the generation. This enforces using more advanced techniques like prompt learning to overcome these ambiguities.

## B.3 Details of Model Hyper Parameters

**Models and training hyper-parameters.** In our experiments, we follow the same setup as ITI-GEN [16] and utilize Stable Diffusion v1.4 [1] as our T2I generator. Then when implementing ITI-GEN we utilize an $S_i$ with a token length of 3 per tSA which is optimized based on a learning rate of $lr = 0.01$. For reference datasets, we utilize [16] readily available datasets with contain 200 reference images per category of each tSA. An Adam [48] optimizer is utilized during prompt learning. For sample generation, we follow the recommended diffusion steps of $l = 50$ and utilize an Attention scale of $c = 10$ and an Attention Queuing transitioning step =10.

**Hard Prompt.** Tab. 6 illustrates the list of HPs utilized in our experiments. We remark that not all tSAs have a clear HP. For example, with the tSA=Skin Colour it is difficult to find a reasonable HP to describe an individual with a specific skin tone.

Table 6: Hard Prompts utilized in T2I generation.

| tSA | Positive Prompts | Negative Prompts |
|---|---|---|
| Male | ''A headshot of a person Male'' | ''A headshot of a person Female'' |
| Young | ''A headshot of a person Young'' | ''A headshot of a person Old'' |
| Smiling | ''A headshot of a person with Smiling'' | ''A headshot of a person with no/without Smiling'' |
| High Cheekbones | ''A headshot of a person with high cheekbones'' | ''A headshot of a person with low cheekbones'' |
| Pale Skin | ''A headshot of a person with pale skin'' | ''A headshot of a person with no/without pale skin '' |
| Eyeglasses | ''A headshot of a person with eyeglasses'' | ''A headshot of a person with no/without eyeglasses'' |
| Mustache | ''A headshot of a person with mustache'' | ''A headshot of a person without mustache'' |
| Gray Hair | ''A headshot of a person with gray hair'' | ''A headshot of a person without gray hair'' |
| Chubby | ''A headshot of a person chubby'' | ''A headshot of a person no chubby'' |

**Stable Diffusion Version.** We verified that the problems of poor performance with Hard Prompts as indicated by Zhang et al. [16] would still persist even in more recent versions of Stable Diffusion e.g., SD 3.0. This issue can be observed by simply inputting the prompt "A headshot of a person without glasses" to SD 3.0 where the generated samples still frequently have the wrong tSA class ("with glasses") indicating this ambiguity still exists. For example, when utilizing the same setup as Tab 1 with HP (from Tab. 6) and SD3.0, our results show poor fairness performance for Eyeglasses (FD=$670e^{-3}$) and Pale Skin (FD=$580e^{-3}$).

**Increasing the size of the reference dataset (2k reference images per category per tSA).** We verified that increasing the size of the reference dataset does not resolve ITI-GEN's quality degradation. Specifically, we repeated the experiment in Tab. 1 with tSA Smiling with 2k reference samples per class. Our results measured an FD=$127e^{-3}$, TA=0.591, FID=89.2, and DS=0.532 which is similar to the results in Tab. 1 (based on 200 reference images). This indicates that the core problem may not be the data size, and may need specific data curation (including sample pairs with only semantic differences in tSA, and similar semantics elsewhere), which poses scalability and applicability challenges.

## B.4 Computation Resources

Tab. 7 illustrates the amount of the compute including the GPU Hours for different steps of our research and the estimated carbon emission for our experiments.

Table 7: **Estimated Computation time**. The carbon emission values are computed using https://mlco2.github.io/impact.

| Experiment | Hardware | GPU Hours | Carbon emitted (kg) |
|---|---|---|---|
| T2I Sample Generation | RTX3090 | 25.0 | 4.87 |
| Directional Alignment analysis | RTX3090 | 0.25 | 0.048 |
| Cross Attention Analysis | RTX3090 | 1.0 | 0.195 |
| Prompt Learning | RTX3090 | 1.0 | 0.195 |
| **Total:** | | 27.25 | 5.308 |

## B.5   Details of the Evaluation Metrics

In this section, we provide more detail on the evaluation metrics used in our work.

**Fairness.** We use the *fairness discrepancy* (FD) metric which compares tSA distribution in generated images with an ideal uniform distribution [27]. In this metric FD=0 would indicate perfect fairness. Following [16, 11, 36] we use a combination of CLIP [20] models, human evaluators, and off-the-shelf models [3] for predicting the distribution of the tSA in target images. We remark that to evaluate FD we utilize classifiers that achieve reasonably high accuracy when evaluated on CelebA [29] reference dataset, as seen in Tab. 8.

Table 8: Accuracy of the tSA classifier when evaluated with CLIP/off-shelf classifier.

| tSA | Accuracy |
|---|---|
| Male | 99.6% |
| Young | 98.8% |
| Smiling | 98.6% |
| High Cheekbones | 97.2% |
| Pale Skin | 98.4% |
| Eyeglasses | 96.7% |
| Mustache | 87.4% |
| Gray Hair | 88.3% |
| Chubby | 90.4% |

**Quality.** Text alignment (TA) [37, 22] and Fréchet Inception Distance (FID) [38] are utilized as quality metrics. Specifically, *Text-alignment* is based on the notation that a sample generated based on ITI-GEN prompt ($P$) or HP ($F$) for a given tSA, should retain the semantics of the original base prompt $T$, unless the sample has degraded in quality. For example, samples generated based on $F = E_T($"A headshot of a person young"$)$ should still retain the semantic of $T = E_T($"A headshot of a person"$)$. Therefore, to evaluate quality, we compare the generated images (based on the $P$ or $F$) with base prompt $T$ using text-image cosine similarity in CLIP's feature space [37, 22]. FID [38] then compares the feature statistics of the generated samples against a reference fair FFHQ [49] dataset from the CleanFID [50] library.

**Semantic Preservation:** When a fairness approach enforces fairness *w.r.t.* a tSA, changes related to that tSA are more favorable. Considering our setup, for the same latent code $Z$, the generated images with base prompt and fairness scheme are more favorable to be different only in features related to tSA, and similar in other features. To measure this behavior, for each input latent code $Z$, the generated image by querying the T2I model with base prompt $T$ is considered as the reference image. Then, for the same latent code $Z$, we measure the similarity of the generated image by querying the T2I with HP and ITI-GEN prompts with that reference image using *DreamSim* [39] features. DreamSim improves upon existing semantic measurement by considering both low-level features, mid-level, and high-level features for a more holistic semantic measurement.

---

[3]https://trust.is.tue.mpg.de/; https://github.com/dchen236/FairFace; https://github.com/SonyResearch/apparent_skincolor

### B.6 Visualizing the Learned Embedding vs Base Prompt

In addition to the direction loss values provided in the main paper to show that $\Delta P$ is not aligned well with $\Delta T$ as pseudo-gold standard direction–which increases the possibility of encoding unrelated knowledge and leading to distorted learned tokens. Here, we provide an additional visualization of the embedding in Fig. 43 to show the difference between the learned embedding in ITI-GEN and the embeddings of the base prompt and hard prompt. As one can see, even though the embeddings of the base prompt and three different variants of the hard prompt are clustered well, the embeddings of the learned tokens in ITI-GEN are quite far from this cluster increasing the possibility of the encoded unrelated concepts and degrading the generation quality. One may argue that using a regularizer can push these embeddings towards this cluster. However, our experimental results suggested that this will result in a compensated performance in terms of the fair T2I generation.

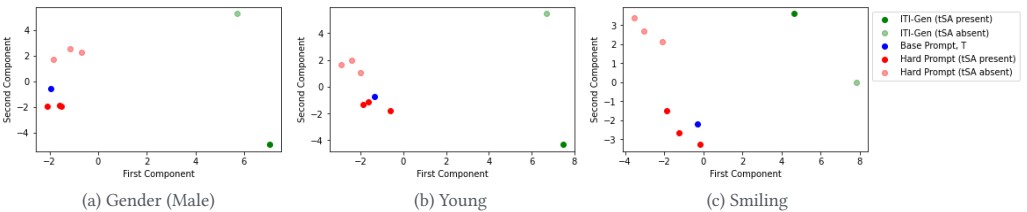

(a) Gender (Male)  (b) Young  (c) Smiling

Figure 43: **PCA Analysis on CLIP-text embedding for ITI-GEN , well-defined HP and Base Prompt.** Utilizing a pre-trained CLIP text encoder we attain the text-embedding for the i) Base Prompt $T=$"A headshot of a person", ii) ITI-GEN tokens, and iii) selected well-defined Hard Prompts for SA∈{Gender (Male), Young}. Then we apply Principle Component Analysis (PCA) for dimensional reduction. We remark that these same text embeddings are later used in the SDM for sample generation.

## C  Limitations and Broader Impact

In this section, we discuss some limitations regarding our work as well as some potential societal impacts that it may have.

**Limitations.** Firstly, FairQueue work follows ITI-GEN which utilizes a reference dataset to first optimize a ITI-GEN prompt $P$. This setup requires dozens of reference images for each category of the tSA which may not always be readily available. Second, although FairQueue provides better quality and semantic preservation of the original base prompt sample, its sample generation is still prone to experiencing entanglement in certain tSA. Specifically, entanglement in tSA could result in some unwanted augmentation of non-tSA during sample generation.

**Broader impact.** Our work FairQueue takes a significant step towards enhancing fairness in text-to-image generation. By improving the quality of samples generated through a fair text-to-image algorithm, we facilitate greater adoption of these techniques by the general public. This increased adoption can help prevent the perpetuation of unwanted biases in everyday applications, promoting a more equitable and inclusive use of technology in society.

# D   Related Work

**Text-to-Image Generation.** There was a surge in text-to-image (T2I) models in the last years, exemplified by models like DALL-E [7, 51], Stable Diffusion [1], Midjourney [52] (with over 16 million active users in July 2023), and many others [53, 24, 54]. These models have demonstrated their capability to generate high-quality samples across different domains and new applications are defined around these models in different areas such as art [55], design, and even medical imaging [56]. In contrast to the earlier generative models which primarily served research purposes within research and scientific settings, current T2I models offer much broader accessibility [57]. However, this increased accessibility also amplifies the potential consequences of bias within these generative models [27]. Our work aims to address this issue by mitigating bias and prompting fair T2I generation.

**Fairness in Generative Models.** In generative modeling, fairness is usually defined as *equal representation* where all categories of a tSA are supposed to be represented with a similar probability. Different approaches are proposed to improve the fairness of the conventional generative models including weak supervision to achieve fairness with the importance weighting of a fair dataset [28], transfer learning from a large biased dataset to a small fair dataset [30, 31], or enforce uniform sampling from the latent feature space [58]. In the context of fair T2I generation, using the pretrained T2I models and adapting prompts for a fair generation has recently attracted a lot of attention. Bansal et al. [17] proposes the use of "ethical interventio" prompts for text-to-image generators to encourage the concept of independence w.r.t. the the sensitive attributes. Specifically, these ethical intervention prompts can be appended to the original prompt *e.g.,* "a photo of a bride *from diverse cultures*" and "a photo of a person wearing a hat *irrespective of their [SA]*" Furthermore, the work substantiates this approach by mentioning that these neutral language prompts exist in the training data. Overall, these additional prompts have empirically been shown to produce diverse samples (a different definition of diversity *w.r.t.* SA) and high-quality samples (human-voted). Chuang et al. [36] proposes to project out of the biased direction of the text embeddings as a form for bias mitigation. Specifically, given a known sensitive attribute, we are able to minimize the equalization loss and find a text embedding that is of equal distance from the biased prompt's embedding. Then utilizing this optimized (debiased) text-embedding, we are able to generate samples with a fairer SA distribution. In addition, recently prompt learning has been proposed to learn the tSA knowledge from a set of reference images in ITI-GEN [16]. In this work, we address the issues that arise due to unrelated concepts being encoded in these prompts, leading to defects in the cross-attention maps and degrading the image generation quality.

**Prompt Learning.** Prompt learning has recently been shown to be a very effective and efficient way of adapting pretrained large vision-language models to different downstream tasks like zero-shot classification [34, 33, 59], image editing [21], personalization of diffusion models [22, 60–62], and few-shot image generation [63]. In this work, we analyze the potential issue of prompt learning in the context of fair T2I generation by analyzing the cross-attention module in the presence of the learned prompts and propose a simple yet efficient approach to integrate prompt learning more efficiently for fair T2I generation.

