# OpenReview forum: "FairQueue: Rethinking Prompt Learning for Fair Text-to-Image Generation"
_NeurIPS.cc/2024/Conference — NeurIPS 2024 poster_

### Official Review · Reviewer_9bUQ · 2024-07-10

**Soundness:** 3
**Presentation:** 3
**Contribution:** 2
**Rating:** 5
**Confidence:** 5

**Summary:**

This paper examines the prompt learning method used in fair text-to-image (T2I) generation, highlighting its impact on image quality. The authors identify that aligning prompt embeddings with reference image embeddings introduces noise due to unrelated concepts, leading to degraded image quality. They conduct an in-depth analysis of the T2I model's denoising subnetwork, introducing novel prompt switching analyses (I2H and H2I) and new quantitative metrics for cross-attention map characterization. Their findings reveal abnormalities in the early denoising steps, which affect global structure synthesis.

To address these issues, the paper proposes two solutions: Prompt Queuing, which applies base prompts initially and ITI-GEN prompts later in the denoising process, and Attention Amplification, which balances quality and fairness. Extensive experiments demonstrate that these methods improve image quality while maintaining competitive fairness and diversity. The paper's contributions include identifying issues in the current prompt learning approach, providing a detailed analysis of the denoising process, and proposing effective solutions to enhance T2I generation quality.

**Strengths:**

1. The paper embarks on an in-depth examination of the denoising subnetwork within the T2I model. The goal is to understand how learned prompts influence image generation throughout the denoising steps. This involves dissecting the entire process from noisy input to the final image, which allows for pinpointing where and how the learned prompts might be causing degradation in image quality. This deep dive includes analyzing how noise is progressively reduced and how the model’s intermediate representations evolve at each step. By doing so, the authors can track the influence of different prompt tokens on the generation process, providing a granular view of the internal dynamics of the model.

2. The cross-attention maps are visual tools that show how attention is distributed across different parts of the image in response to various tokens in the prompt. By mapping out these attention distributions, the paper identifies specific patterns and irregularities. The analysis reveals that certain tokens, which are not directly related to the target sensitive attributes (tSAs), exhibit abnormally high levels of activity. For instance, non-tSA tokens like "of" and "a" show higher attention scores, indicating that the model's focus is scattered and not properly aligned with the relevant attributes. This misalignment disrupts the formation of coherent global structures in the early stages of image generation.

3. The paper highlights that abnormalities in cross-attention maps are particularly pronounced during the initial steps of the denoising process. At these stages, the model is supposed to establish the foundational global structure of the image. However, the distorted prompts from ITI-GEN interfere with this process, leading to poorly structured outputs. For example, the analysis shows that when using ITI-GEN prompts, the model often fails to correctly focus on key image regions, resulting in artifacts and unintended features. This is contrasted with the behavior of hard prompts, which show more targeted and consistent attention patterns that help maintain image quality.

4. This paper includes extensive experimental results demonstrating the superiority of FairQueue over ITI-GEN in terms of fairness, image quality, and semantic preservation. These  experiments are conducted on multiple datasets and with various tSAs.

**Weaknesses:**

1. The proposed innovations, Prompt Queuing and Attention Amplification, are relatively incremental and might be seen as a combination of existing techniques rather than a breakthrough innovation. The combination of these two mechanisms does improve performance but lacks a strong theoretical underpinning to justify why this specific combination is superior.
2. The proposed method, FairQueue, while innovative, could benefit from a more detailed explanation of how it addresses the issues identified in the paper. In the current form, the explanation of Prompt Queuing lacks depth regarding why the base prompt T is more effective in the early denoising steps. The authors should delve deeper into the reasons behind this, possibly by providing theoretical analysis and specific experimental results that demonstrate how the base prompt gradually establishes the fundamental elements of the image during the denoising process.
3. Additionally, the transition to ITI-GEN prompts in the later stages should be better explained, particularly in terms of how it enhances the detailed expression of tSAs. Comparative illustrations of image generation results at different stages using different prompts would effectively show the benefits of this transition.
4. The mechanism of Attention Amplification needs a clearer explanation. The authors should detail why scaling the cross-attention maps enhances tSA expression, possibly through mathematical derivations or experimental data showing the impact of different scaling factors on image quality and tSA expression.
5. Although the experimental results are comprehensive, the paper lacks detailed descriptions of the experimental settings and specific parameters used, which could hinder reproducibility.

**Questions:**

1. The authors provide a more detailed explanation or evidence on why the base prompt T is more effective in forming global structures during the early denoising steps compared to ITI-GEN prompts. Including theoretical justifications or experimental comparisons that illustrate how the base prompt helps establish the foundational elements of the image more effectively would be beneficial. Specific examples or visualizations from your experiments could clarify this point.
2. How does switching to ITI-GEN prompts in the later stages of denoising improve the detailed expression of tSAs without compromising the overall image quality?
3. The author should elaborate on the mechanism by which attention amplification enhances tSA expression. How is the scaling factor chosen and validated?
4. The paper would be strengthened by more detailed experiments and results analysis, especially comparisons between using only the base prompt, only ITI-GEN prompts, and the FairQueue method. A thorough analysis of these results would help demonstrate how FairQueue effectively addresses the issues identified with the original method. By incorporating these detailed explanations and analyses, the paper would more convincingly show how FairQueue resolves the problems it aims to address, enhancing its credibility and scientific rigor.

**Limitations:**

1. Although the authors have identified and analyzed the issues with the current methods, the explanation of how their proposed method, FairQueue, effectively addresses these issues lacks detail. Regarding the effectiveness of Prompt Queuing, the authors should provide a deeper explanation of why the base prompt is more effective in the early denoising steps. This could be supported by theoretical analysis and specific experimental results, such as demonstrating how the base prompt gradually establishes the fundamental elements of the image during the denoising process. Further explanation of how using the base prompt in the early steps helps form a robust global structure, allowing the application of ITI-GEN prompts in later steps to better add details without compromising the overall structure, would be beneficial.
2. For the mechanism of Attention Amplification, the authors need to elaborate on why scaling the cross-attention maps effectively enhances tSA expression. This section could include mathematical derivations or experimental data to show how different scaling factors impact image quality and tSA expression. Providing specific examples of how attention amplification compensates for the reduced tSA expression due to prompt queuing in various scenarios would strengthen the argument.
3. The authors should include more experimental results, particularly comparisons between using only the base prompt, only ITI-GEN prompts, and the FairQueue method. Detailed analysis of these results would help demonstrate how FairQueue addresses the issues identified with the original method, such as image quality degradation and insufficient tSA expression. Providing specific quantitative metrics and visual comparisons can more intuitively show the advantages of the proposed method.
4. The paper lacks a thorough analysis of the computational complexity and the necessary resources for implementing FairQueue. Understanding the computational demands is essential for assessing how feasible and scalable the method is in real-world applications. Without this information, it is difficult to determine whether FairQueue can be efficiently deployed in various environments, particularly those with limited computational resources.

---

> ### Author Rebuttal · Authors · 2024-08-07
>
> We thank Reviewer for detailed feedback. Due to lack of space we consolidate the comments from Weakness(W), Questions(Q), and Limitations (L) into 7 responses.
>
> $ $
>
>  >R4Q1: “proposed innovations, Prompt Queuing, and Attention Amplifcation are relatively incremental” (W1)
>
> R4A1: We respectfully remark **our proposed solution for quality degradation in sophisticated fair generative model is non-trivial**, requiring our deep analysis of attention maps in individual denoising steps via our novel H2I/I2H analysis:
>
> - I2H reveals **learned tokens (via ITI-GEN) affect early denoising steps, degrading global structure**
> - H2I reveals **learned tokens work well in enhancing tSA expression in the later denoising steps, if global structure synthesized properly**
>
> *only from these observations*, can we propose FariQueue which is both simple and effective in mitigating quality degradations.
>
> $ $
>
> >R4Q2: “ why the base prompt $T$ is more effective in the early denoising steps” (W1,W2,Q1,Q4,L1,L3)
>
> R4A2: We apologize if emphasized insufficiently that the effectiveness of base prompt $T$ is grounded on our I2H and H2I analysis. Recall that I2H reveals quality degradation is related to using learned tokens (via ITI-GEN) in early denoising stage, and H2I reveals that using HP (no learned tokens) instead can prevent it.  **Using $T$ is in fact grounded on this analysis: both $T$ and HP are free of learned tokens.**  This is further seen in Supp B.6 where $T$ and HP are close in the embedding space while the learned prompts are far. (Recall that HP can only be used for a few tSA with minimal linguistic ambiguity, thus using $T$ is needed as a solution for many tSA).
>
> In addition, per Reviewer’s comment, we provide visualizations of $T$’s effectiveness in generating the global structure in early denoising steps. Specifically, we compare the cross-attention maps of FairQueue with ITI-Gen during sample generation, together with quantitative analysis. Results in col 2 vs 3 of Fig. A (rebuttal pdf) illustrate $T$’s effectiveness in synthesizing the global structure in stage 1, and non-abnormal attention (in Fig B), resulting in effective global synthesis than ITI-Gen and better sample quality.
>
> $ $
>
> >R4Q3: “How does switching to ITI-GEN prompts in the later stages of denoising improve the detailed expression of tSAs without compromising the overall image quality” (W1, W3, Q2, Q4, L1, L3)
>
>
> R4A3: As our I2H/H2I analysis suggests, given good global structures synthesized in stage 1, ITI-Gen tokens enforce the tSA in stage 2 by having good *localized* attention that attends well to *tSA-specific regions* resulting in gradual addition of detail that enhances tSA expression.
>
> Also, per Reviewer’s comment, we provide visualization for  FairQueue vs $T$, where they are the same in stage 1 but are different in stage 2 (only FairQueue contains learned tSA tokens).
>
> Results in col 1 vs 2 of Fig. A (rebuttal pdf) show that with FairQueue, the learned tokens $S_i$ emphasize attending to tSA-related regions (e.g., eyes and mouth for Smiling, or lower half of the face and cheeks for High Cheekbones),  to output samples with the tSA features. Furthermore, as the attention is generally localized to the tSA-specific regions, the sample’s quality is well preserved.
>
> $ $
>
> >R4Q4: “Elaborate on the mechanism by which attention amplification enhances tSA expression. How is the scaling factor chosen and validated” (W1, W4, Q3, L2)
>
> R4A4: As discussed in R4A3, learned ITI-Gen tokens enforce tSA expressions by attending to tSA-specific regions. Attention Amplification then increases this tSA expression by amplifying the attendance at tSA-specific regions.
>
> Then, an ablation study in Supp A.2 determined the optimal scaling factor $c$. Our results suggest that $c=10$ delivers Pareto optimal performance for fairness and quality.
>
> Additionally, per Reviewer’s comment, we provide some qualitative results (Fig C. rebuttal pdf) to support this analysis. Our results show that at $c=0$, even with prompt queuing, samples may lack tSA expression (Smiling). Increasing  $c$ helps to enhance tSA expression.
>
> $ $
>
> >R4Q5: Justify why this specific combination (PromptQueing and Attention Amplifcatio n) is superior. (W1)
>
> R4A5: To verify the contribution of both PromptQueuing and Attention Amplification, our ablation study in Supp A.2. varied the amplification scaling factor, $c$, and Prompt Queuing transition point.
>
> Our analysis reveals that both components contribute to FairQueue's superior performance and **removing any one results in worsened performance**, i.e., when $c=0$, FD increases (less fair), and when the transition point=0, TA decreases (poorer quality).
>
> $ $
>
>  >R4Q6: “The paper lacks detailed descriptions of the experimental settings and specific parameters used …” (W5)
>
> R4A6: We respectfully clarify that all experimental reproducibility details are already available in Supp:
> - **Model + hyperparameter** in Supp B.3
> - **Code + reproduction files** in the anon. link (line 488)
> - **Evaluation metrics**  in Supp B.5
>
> Furthermore, all reproducibility resources are in the Supp.
>
> $ $
>
>  >R4Q7: “The paper lacks a thorough analysis of computational complexity.” (L4)
>
> R4A7: We clarify that we had conducted a thorough analysis to verify that FairQueue contributes **negligible complexity**.
> For inference: the inference time of the generation is ~45s regardless of base prompt, ITI-Gen, or FairQueue. It is because PromptQueing adds negligible computation to replace $T$ with $P$, and Attention Amplification only introduces a multiplication operation. General computation resources info is provided in Supp B.4, and we will add more details to clarify this.
> For prompt learning: prompt learning (ITI-Gen/FairQueue) is generally light-weight due to:
>
> - T2I generator is frozen and only three tSA tokens are trained.
> - T2I generator is not in the training loop
>
> On average, learning tokens for a tSA takes ~5min with one RTX 3090.

---

> > ### Comment · Reviewer_9bUQ · 2024-08-13
> > **Response for authors**
> >
> > Thank you for the rebuttal. I still have one concern. The authors conducted ablation studies to validate the contribution of each component, which is well-handled. However, the explanation of why the specific combination of Prompt Queuing and Attention Amplification is superior to other possible combinations could be more detailed. A deeper analysis of why this particular combination works best would strengthen your argument.

---

> ### Author Response · Authors · 2024-08-13
> **Clarifying FairQueue’s as the Best Performing Combination**
>
> >R4Q8: Explanation of why the specific combination of Prompt Queuing and Attention Amplification is superior to other possible combinations could be more detailed.  Deeper analysis of why this particular combination works best would strengthen your argument
>
> We thank Reviewer 9buQ for insightful comments. Here we provide analysis of all possible combinations of Prompt Queuing (PQ) and Attention Amplification (AA). We note that all these analyses are already in the paper to derive the best combination, but we fully agree with Reviewer that it is better to have them in one place, to clearly support why our PQ+AA is superior.
>
> Recall:
>
> 1. PQ enables proper global structure generation, leading to **improved sample quality**.
> 2. AA supplements PQ by enhancing exposure to tSA tokens, leading to **state-of-the-art fairness**.
>
> With these, we provide analysis of why PQ+AA works better than all other combinations.
>
> $ $
>
> Tab i: Analysis of **all possible different combinations** for Prompt Queuing (PQ) and tSA Attention Amplification (AA). We summarize our findings from main paper for the tSA “Smiling”. Note $\alpha (S)$ notates AA for tSA tokens and results in **bold** and *italics* are the best and second best. Notice that C6:FairQueue (PQ+AA) provides the best combination: it achieves **both** outstanding sample quality (C6: TA=0.674 & FID=80.02 similar to C1: TA=0.681 & FID=76.9 with the best quality but poor fairness) and fairness (C6: FD=0.069 similar to C4: FD=0.05 with the best fairness but poor quality).
>
>
>
> |  | Prompt Queueing (PQ) | Attention Amplification (AA) | Stage 1 Prompt | Stage 2 Prompt | FD$\downarrow$ | TA$\uparrow$ | FID$\downarrow$ | DS$\downarrow$ | Remarks |
> |---|---|---|---|---|---|---|---|---|---|
> | C1: **no PQ, no AA** for Base Prompt | No | No | $T$ | $T$ | 0.211 | **0.681** | **76.9** | - |  |
> | C2: **no PQ, no AA** for ITI-Gen | No | No | $[T;S]$ | $[T;S]$ | 0.124 | 0.605 | 88.63 | 0.557 |  |
> | C3: **AA only** for Base Prompt | No | Yes | $T$ | $T$ | N.A | N.A | N.A | N.A | An **unimplementable combination** due to the absence of tSA tokens for AA. |
> | C4: **AA only** for ITI-Gen | No | Yes | $[T;S]$ | $[T;\alpha (S)]$ | **0.05** | 0.61 | 89.41 | 0.55 |  |
> | C5: **PQ only** | Yes | No | $T$ | $[T;S]$ | 0.145 | *0.674* | 80.15 | **0.24** |  |
> | C6: **PQ + AA** (Our specific combination) | Yes | Yes | $T$ | $[T;\alpha (S)]$ | *0.069* | *0.674* | *80.02* | *0.284* | Both PQ and AA are present i.e., our proposed FairQueue |
>
> $ $
>
> With the results summarized from main paper (above), we provide deeper analyses below with additional explanations for improved clarity:
>
> - **C1: Base Prompt $T$ Only** (no AA no PQ): It lacks tSA-specific knowledge and results in poor fairness. Additionally, without tSA tokens $S$, AA is not applicable for **C3**.
>
> - **C2: ITI-Gen prompt $P$ Only**, in Tab.1 (no AA no PQ). Our analysis in Sec 3.2. shows it has poor quality due to distortion in global structure during sample generation. Without PQ, the issue of distorted global structure persists for some tSAs.
>
> - **C4: Attention Amplifcation (AA) Only**, in Supp. A.2. Fig.22 when PQ transition point=0. It results in poor quality since only ITI-Gen is used. We remark that utilizing only AA for ITI-Gen may deceptively improve fairness, but the generated samples have poor quality e.g., Smiling cartoons. The reason is (similar to C2): without PQ, the issue of distorted global structure persists for some tSAs.
>
> - **C5: Prompt Queuing (PQ) Only**, in supp. A.2 Fig 22 when $c=0$. By replacing the distorted ITI-Gen prompt with the Base prompt in Stage 1,  PQ leads to improved quality, but without AA, the fairness remains poor given reduced exposure to tSA tokens in the denoising process.
>
> - **C6: FairQueue (PQ+AA)**, in Tab.1. Our proposed solution with optimal quality and fairness. Specifically, it combines the effects of Prompt Queuing– enabling the global structure to be properly formed resulting in good quality samples, and Attention Amplification–enhancing the tSA-specific expression for better fairness.
>
> $ $
>
>
> The results in Tab.i present a summary of related quantitative results:
> - Comparing C4, C5, and C6 reveals that both PQ and AA are necessary to obtain high-quality samples with good fairness performance. Specifically, C4 has poor quality, while C5 has poor fairness.
> - Comparing C1, C2, and C5 reveals the necessity of Prompt Queuing because utilizing either only ITI-Gen or Base Prompt results in quality and fairness degradation, respectively.
>
> Overall, our quantitative analysis demonstrates **C6:FairQueue** is the superior combination, balancing both fairness and quality. It achieves a TA=0.674 and FID=80.02, the closest to the best quality by C1 with TA=0.681 and FID=76.9 (but poor fairness), while achieving fairness of FD=0.069, the closest to the best fairness by C4 with FD=0.05 (but poor quality).
>
> We hope this further discussion addresses Reviewer’s concerns.

---

> > ### Author Response · Authors · 2024-08-14
> > **Appreciation for the Discussion**
> >
> > As we will conclude the Rebuttal’s discussion session in 5 hours, we sincerely thank the Reviewer 9buQ for their valuable insights and thoughtful discussion. We hope that our efforts to address the Reviewer’s comments have been satisfactory.
> >
> >
> > *If the Reviewer finds our responses appropriate, we kindly ask for a re-consideration of our scores based on the new additional understanding of our work.*

---

### Official Review · Reviewer_iQtH · 2024-07-12

**Soundness:** 3
**Presentation:** 3
**Contribution:** 3
**Rating:** 7
**Confidence:** 3

**Summary:**

The authors propose FairQueue, a simple and effective solution to solve the quality degradation problem in ITI_GEN through prompt queuing and attention amplification.

**Strengths:**

1.	The paper is well-written and logically structured.
2.	The proposed FairQueue effectively solves the quality degradation problem in ITI_GEN.
3.	The paper is very rich in experiments

**Weaknesses:**

Some of the contribution points of the article can be optimized.

**Questions:**

1.	In Chapter 4, the author proposed a method for attention amplification. Is the amplified area Si related to tSA? If so, please explain the specific impact relationship.
2.	Are there specific metrics to measure the relationship between generation quality and fairness?

**Limitations:**

The author can further improve the description of the method part of the paper and describe the innovation more clearly.

---

> ### Author Rebuttal · Authors · 2024-08-07
>
> >R3Q1: Some of the contribution points of the article can be optimized.
>
> R3A1: We appreciate the Reviewer’s comment and would like to assure the Reviewer that we have already optimized our contributions with further discussion in the supplementary (due to space limitations). Specifically,
>
> - In Supp. A.1., to optimize the delivery of our findings in cross-attention analysis, we provide substantial qualitative samples on different tSAs to demonstrate the characteristics of natural language and ITI-Gen prompts. Specifically, ITI-Gen prompts affect the global structure in the early denoising steps, while effectively enforcing the tSA once the global structure has been well formed by natural language prompts.
>
> - In Supp. A.2, we present an ablation study to optimize the effect of both mechanisms in our proposed method: Prompt Queuing and Attention Amplification. Here, we observe that each component contributes to improving either fairness or quality of the generated sample.
>
> - In Supp B.6, to advance our understanding of the distorted ITI-Gen prompts, we provide further discussion on the issues with ITI-Gen prompts. Here, we provide additional analysis to show that ITI-Gen’s prompt embeddings deviate significantly from the Base prompt and Hard prompt embedding. These findings further support our understanding of the differences between ITI-Gen prompts and natural language prompts (HP and Base prompt).
>
> We hope that these additional resources address the reviewer’s comment. We will include additional pointers to them.
>
>
> $ $
>
>
> >R3Q2: Is the amplified area Si related to tSA? If so, please explain the specific impact relationship.
>
> R3A2: We remark that the Reviewer’s understanding of the amplified area is correct. Specifically, the $S_i$ tokens in the ITI-Gen learn to encode the tSA by minimizing the directional loss (as discussed in lines 108-118). Therefore, the attention map correlating to the $S_i$ tokens is responsible for expressing the tSA at related regions, so is the amplified $S_i$.
>
> Particularly, recall that prompt-queuing allows for the global structure to be first generated with base prompt $T$ followed by tSA adaptation with learned prompt $P$. This reduces the number of denoising steps that generated samples are exposed to the $S_i$ tokens, i.e., only in stage 2 and hence affecting tSA expression. To compensate for this, Attention Amplification (discussed in lines 267-270) was introduced, which scales the attention maps correlating to these $S_i$ (the tSA-specific tokens). This results in the intensification of the tSA-specific regions (e.g., attending with more intensity to the mouth region in tSA=’Smiling’), and therefore enhances the tSA expression in the generated samples.
>
> To further understand the impact of attention amplification, we direct the Reviewer to our ablation study in Supp. A.2, where we augment the attention amplification factor $c$. Here, we observe that when $c=0$, the FD is still relatively large due to the lack of tSA expression. Then, by increasing this to $c=10$, we observe the increment in the tSA expression lowers the FD. To further support this point, in this rebuttal, we provide some additional qualitative results seen in Fig. C of the attached pdf. Here, we observe the same effect on ‘Smiling’ where the tSA expression is poor at $c=0$. Then when amplifying the attention intensity ($c>0$) to the related regions (e.g., mouth, as seen in rebbutal Fig. A col 2–FairQueue) we observe that the tSA expression also intensifies i.e., samples begin to smile.
>
>
> $ $
>
>
> >R3Q3: Are there specific metrics to measure the relationship between generation quality and fairness?
>
> R3A3: We thank the Reviewer for the insights and agree with the Reviewer’s intuition that there should exist a metric to evaluate the relationship between fairness and quality since this trade-off can exist. For example, some existing fair generative models in the literature [27, 28, 30]  require balanced datasets for the tSA, which are commonly small due to their difficulty to collect, and therefore affect sample quality. Unfortunately, to the best of our knowledge, the fair generative modeling community has not studied such a metric yet. Therefore, in our manuscript, following related work [16,28,30,31] we consider existing well-known individual metrics in the literature to assess fairness and quality, separately. Specifically as mentioned in line 140 and discussed in more detail in Supp. B.4, we have followed the existing works in the literature [16, 28, 30, 31], and used the following metrics for evaluating the fairness and quality of the generated images:
>
> - Fairness discrepancy (FD) is a common metric utilized in many popular fair generative modeling works [16,27,28,30,31] to assess fairness in the context of sample generation.
>
> - FID [16,38,28,30,31,35,60] is a very popular quality metric used in generative modeling to compare the feature embedding distribution on the generated sample against an ideal reference dataset.
>
> - Text Alignment [16,37,11,60] is a popular quality metric used in evaluating text-to-image models.
>
> - DreamSIM [39,B,C] extends on the well-known LPIPS [35,60] metric in the literature, to determine semantic similarities but additionally considers the high-level, mid-level, and low-level semantics concurrently.
>
>
> $ $
>
>
> >R3Q4: The author can further improve the description of the method part of the paper and describe the innovation more clearly.
>
> R3A4: We appreciate the reviewer's feedback and hope that our responses have clarified the details of the method in our paper. In the final manuscript, we will include this content from the Supp. We remark that all our code and necessary instructions are included in the Supp for full reproducibility of our work.
>
> $ $
>
> **We appreciate the positive feedback and valuable comments of the reviewer. we sincerely hope that reviewers could consider increasing the ratings if our responses have addressed all their questions**.

---

### Official Review · Reviewer_8Wzu · 2024-07-12

**Soundness:** 3
**Presentation:** 1
**Contribution:** 3
**Rating:** 5
**Confidence:** 1

**Summary:**

This paper introduces an approach to improving fairness and quality in text-to-image generation models by rethinking prompt learning. The current Inclusive Text-to-Image Generation (ITI-GEN) methods often degrade image quality due to suboptimal training objectives and embedding misalignment between learned prompts and reference images.
Key innovations include Prompt Queuing, ensuring regular and fair prompt usage, and Attention Amplification, enhancing the impact of prompts on image generation. Experiments show that FairQueue improves both image quality and fairness compared to state-of-the-art methods.

**Strengths:**

- Through detailed analysis, they revealed the limitations of the previous approaches, specifically how suboptimal training objectives and embedding misalignments degrade image quality.

**Weaknesses:**

- The readability of the paper is poor.
- The evaluation of the model is insufficient, as there are only evaluations on the datasets of human faces, which lacks in evaluating the generalizability of the model. More experiments on other datasets are needed.
- The paper emphasizes quantitative metrics but does not give enough attention to qualitative assessments. Including user studies or expert reviews could provide valuable insights into the perceived quality and fairness of the generated images, complementing the quantitative data.

**Questions:**

- I'm curious why the experiments were conducted on only one dataset.
- Some of the metrics used in the paper are unfamiliar. Are these metrics validated, and are they commonly used in other research papers as well?
- Additionally, I would like to know how you plan to address the limitations mentioned in the limitations section.

**Limitations:**

They mentioned limitations and broader impact in the paper.

---

> ### Author Rebuttal · Authors · 2024-08-07
>
> >R2Q1:The readability of the paper is poor.
>
> R2A1: We remark that Reviewer 89En scored our presentation excellent, and Reviewer iQtH described it as “well written and logically structured.” We will review the paper once again.
>
> $ $
>
> > R2Q2: I'm curious why the experiments were conducted on only one dataset.
>
> R2A2: We respectfully clarify that following ITI-Gen paper [16], our work conducts experiments on **3 datasets** including CelebA [29], FairFace [45] and Fair Benchmark [44] (line 277).
>
> $ $
>
> >R2Q3: The evaluation of the model is insufficient, as there are only evaluations on the datasets of human faces, which lacks in evaluating the generalizability of the model.
>
> R2A3: We remark that for the type of datasets, we followed literature [16, 28, 30] on fairness. Specifically, as the literature is mostly focused on the human-centric tSA (e.g., “Gender”, “SkinTone”, …) expressed by human faces, existing fairness datasets are developed and curated based on this concept.
>
> We remark that our approach is **agnostic to the dataset type** with no restriction regarding the type of content. However, to the best of our knowledge, the fairness community has not developed a curated dataset for this purpose. As an example, the FairFace dataset is curated for fairness study to include 6 skin tone classes and considers the Individual Typology Angle [44] to ensure that race is uncorrelated with skin tone. However, a similar effort has not been carried out for non-human concepts in fairness literature to enable studying fairness in these concepts.
>
> $ $
>
> > R2Q4: The paper emphasizes quantitative metrics but does not give enough attention to qualitative assessments.
>
> R2A4: We thank the Reviewer for the comment. First, we clarify that we have tried to provide extensive qualitative assessment between ITI-Gen and FairQueue samples in [Annon. Supp link line 489]  \> more resources \> Tab_1 (due to size limitation) containing 1.8k samples per approach. Nevertheless, we follow Reviewer’s suggestion and include a user study experiment to compare ITI-Gen and FairQueue. Specifically, we utilize the same seed to generate 100 sample pairs with ITI-Gen and FairQueue for ‘Smiling’, ‘High C.’, ‘Gender’, and ‘Young’. Then, utilizing Amazon Mechanical Turk we conduct 2 tasks:
>
> -  **Quality comparison by A/B testing**:  Human labelers select the better quality sample between ITI-Gen and FairQueue (from the same seed). Each sample was given to 3 labelers.
>
> - **Fairness comparison by human-recognized tSA**: labelers identified the tSA class for each sample. The final label was based on the majority of 3 labelers. labelers were also given an “unidentifiable” option if the class could not be determined. Finally, the labels were used to measure FD.
>
> Our results in Tab. A reveals that FairQueue generates better quality samples than ITI-Gen (>62.0% preference) and Tab.B shows that FairQueue achieves competitive fairness with ITI-Gen. Overall, this aligns with our quantitative results in Tab.1.
> We will include these results in the final paper.
>
> $ $
>
> Tab. A: **A/B testing:** Human assessment comparing quality between ITI-Gen vs FairQueue for 200 samples per tSA. Col 2 and 3 indicate the percentage of labelers that prefer the method’s sample quality. A larger value is better.
>
> |  |  ITI-Gen | FairQueue |
> |---|---|---|
> | Smiling | 1.3 % | 98.7 % |
> | High.C | 2.7 % | 97.3 % |
> | Gender | 33.0% | 67.0% |
> | Young | 38.0% | 62.0% |
>
> $ $
>
> Tab.B: **Fairness comparison by human-recognized tSA:** Human assessment to compare FD $\downarrow$ for ITI-Gen vs FairQueue for 200 samples per tSA.
>
> |  |  ITI-Gen FD | FairQueue FD |
> |---|---|---|
> | Smiling | 0.106 | 0.014 |
> | High.C | 0.144 | 0.021 |
> | Gender | 0.014 | 0.014 |
> | Young | 0.014 | 0.028 |
>
>
>
> $ $
>
> >R2Q5: Some of the metrics used in the paper are unfamiliar. Are these metrics validated, and are they commonly used in other research papers as well?
>
> R2A5: We assure the Reviewer that as discussed in line 140, with more details in Supp B.5, these are indeed popular performance metrics. Specifically,
>
> - FD is a common fairness metric utilized in many popular fair generative modeling works [16,27,28,30,31]
> - FID [16,38,28,30,31,35,60] is a very popular quality metric used in generative modeling.
> - Text Alignment [16,37,11,60] is a popular quality metric used in T2I models.
> - DreamSIM [39,B,C] extends on LPIPS [35,60], a popular metric to evaluate semantic similarity of images.
>
> $ $
>
> >R2Q6: Additionally, I would like to know how you plan to address the limitations mentioned in the limitations section.
>
> R2A6: We appreciate the Reviewer's interest in our work. We remark that the limitations section is included with the aim of full transparency. In this paper, as we focus on improving quality of fair generative models, these limitations, which require substantial research effort, are beyond the current scope. Considering these points, in what follows we provide some discussion for potential future works.
>
> We first discuss attribute entanglement, which we believe is largely due to the current setups in fair generative models which lack an ideal dataset that fully disentangles the tSAs. For example, the attribute ‘Bald’ is often associated with ‘Male’ rather than ‘Female’. As a result, the entanglement issue is shared among all fairness approaches and a solution is non-trivial. However, it is indeed a very interesting research direction, which we hope to explore.
>
> Next, we address the need to have a carefully-constructed reference dataset to guide the training of ITI-Gen tokens. In future works, we may take inspiration from the few-shot generation literature like StyleGAN-Nada [35] which utilizes an expert auxiliary model to guide the training in place of reference data.
>
> We will add these details to future work.
>
> $ $
>
> **We appreciate the valuable comments and sincerely hope that reviewers could consider increasing the ratings if our responses have addressed all their questions**.

---

> > ### Comment · Reviewer_8Wzu · 2024-08-12
> >
> > Thank you for the rebuttal. I think the author has addressed all the concerns that I raised. I updated my score to 5.

---

> ### Author Response · Authors · 2024-08-13
> **We appreciate the Reviewer Raising our rating and for the kind words**
>
> We are very thankful for the Reviewer's increased rating and that we were able to “addressed all the concerns that I  (the reviewer) raised”
>
> We respectfully seek the reviewer to consider adjusting their presentation and/or confidence score to reflect their new understanding, should the reviewer find it appropriate.

---

### Official Review · Reviewer_89En · 2024-07-12

**Soundness:** 2
**Presentation:** 4
**Contribution:** 2
**Rating:** 5
**Confidence:** 3

**Summary:**

In the field of fair text-to-image generation, the ITI-GEN paper demonstrated good performance by learning and embedding tSA. However, this paper argues that the embeddings learned in ITI-GEN can include unrelated attributes, resulting in noisy embeddings and significantly degrading the quality of the generated samples. To address this, the paper proposes using prompt queuing to apply tSA after a certain timestep and employs attention amplification to prevent the weakening of tSA.

**Strengths:**

The paper provides a highly detailed analysis of how the learned tSA embeddings are actually reflected in the images using attention maps. It also demonstrates that the learned embeddings do not properly reflect differences in the images.

**Weaknesses:**

First, the degradation seems severe compared to the ITI-GEN paper. If I understand correctly, ITI-GEN appears to aggregate all learned tSA embeddings by summing them, while this paper seems to concatenate a large number of tokens, as indicated by (p+q)*d. This difference in implementation may be causing a decline in baseline performance and requires an explanation.
Additionally, the proposed method to address this issue seems naive, and there is a lack of ablation experiments. Also, even with the proposed method, there appears to be a degradation in sample performance compared to HP. For example, in Figure 2, an originally female image appears to have been changed to a male image using the proposed method.

**Questions:**

- Is there a difference between the original ITI-GEN and your re-implementation as I mentioned or did I just misunderstand? (I am not an expert of ITI-GEN)
- Wouldn't this problem be naturally resolved if a large number of reference images were used to learn the embeddings?

**Limitations:**

The authors describe the limitations and potential societal impact in their supplemental material.

---

> ### Author Rebuttal · Authors · 2024-08-07
>
> > R1Q1: “ITI-GEN appears to aggregate all learned tSA embeddings…This difference in implementation may be causing a decline in baseline performance”
>
> R1A1: We thank the Reviewer for their comment, and apologize if it was unclear. We clarify that our code of  ITI-Gen as a baseline is **100% identical** to the original ITI-Gen paper as we utilize the official GitHub source code released by ITI-Gen authors (line 282). Therefore, **we are certain that the quality problem is sourced from ITI-Gen**.
>
> For multiple tSAs, following ITI-Gen exactly, we used the aggregation mechanism denoted in ITI-Gen’s paper Eqn 3. We will include these details in our paper for further clarification.
>
> We reassure the Reviewer on the above matters by directing them to the submitted source code to observe the existence of this aggregation function: **[Annon. Link line 489] \> iti_gen \> model.py (line 225-230)**.
>
> We hope the provided details address the Reviewer's concerns.
>
> $ $
>
> >R1Q2: the proposed method to address this issue seems naive.
>
> R1A2: We respectfully clarify that proposing a solution for the quality degradation issue is **non-trivial**. Our proposed scheme stems from uncovering this issue in the context of prompt learning for fair T2I generation. **Only through detailed analysis of the issue (highlighted as a core contribution) and examination of attention maps of individual denoising steps, were we able to propose a simple yet effective solution.**
>
> Specifically, first, we uncover the issue of quality degradation in ITI-Gen (based on their code). Then, we attribute this issue to the sub-optimal directional loss used for prompt learning in ITI-Gen leading to encoding unrelated concepts in learned tokens (Fig 1a-col3). We further trace back the effect of these learned tokens in sample generation by analyzing the cross-attention mechanism (Fig. 3), and propose H2I and I2H analysis to analyze the effect of the learned tokens on different stages of the denoising process.
>
> **Only upon this analysis** do we recognize that the distorted ITI-Gen tokens affect the global structure in the early denoising steps, but work well in enforcing tSA expressions if the global structure is formed properly–by natural language prompts like HP and Base Prompt. This inspires us to propose prompt queuing to bypass degrading global structure in the early steps of denoising.
>
> Although Prompt queuing prevents degrading global structure, as there are no tSA-related tokens in the early stage, it results in reduced tSA expression. However, our H2I analysis suggests that with proper global structure, tSA tokens attend to tSA-related regions to enforce fair generation. Inspired by this analysis, we propose attention amplification to enlarge the effect of tSA tokens and intensify tSA expression to achieve fairness without degrading image quality.
>
> $ $
>
> >R1Q3: there is a lack of ablation experiments
>
> R1A3: We respectfully correct the review that ablation studies are included in Supp. A.2 (mentioned in line 297) due to space limitations. In these ablation studies, we experimented with different intervals for prompt queueing and scales for attention amplification. These ablation studies verify the effectiveness of individual components.
> We additionally also compared FairQueue against ITI-Gen in the application of  Training-Once-For-All. Our results again demonstrate FairQueue improved quality over ITI-Gen.
>
> $ $
>
>
> >R1Q4: with the proposed method, there appears to be a degradation in sample performance compared to HP… e.g. in Figure 2, female image appears to have been changed to a male.
>
> R1A4: We clarify that based on the DS values (measured on 1k samples per tSA), our proposed approach on average has better performance in semantic preservation compared to HP, e.g, 0.284 and 0.330 for ‘Smiling’ and ‘High Cheekbones’ compared to 0.323, and 0.332 for HP (DS($\downarrow$) is a metric for measuring semantic preservation, see our paper). We remark that comparing degradation of two approaches based on a single sample might not be that easy, as in Fig. 2, HP also shows some discrepancies similar to what indicated by Reviewer, e.g., in col. 4, row 5, HP changes a male to female.
>
> Importantly, we remark that HP is only applicable to a few tSAs which have minimal linguistic ambiguity (MLA). For most tSA which are non-MLA (see Supp A.4), HP has a poor performance in enforcing fairness (as also discussed in the ITI-Gen paper). In contrast, as shown in Tab. 1, FairQueue delivers a good performance in terms of both quality and fairness across various tSAs.
>
> $ $
>
> > R1Q5: Is there a difference between the original ITI-GEN and your re-implementation
>
> R1A5: As answered in R1A1: Our deployment of ITI-Gen as a baseline is 100% identical to the original work.
>
> $ $
>
>
> >R1Q6: Wouldn't this problem be naturally resolved if a large number of reference images were used to learn the embeddings?
>
> R1A6: We believe that simply increasing the reference data size may not resolve the problem of unrelated knowledge. Specifically,  the existing ITI-Gen reference dataset is already relatively sizable. For example, in CelebA, 200 samples are used for each class of tSA, and yet this problem persists.
>
> To verify this, we conduct the same experiment with tSA Smiling with 2k reference samples per class. Our results measured an FD=$127e^{-3}$, TA=$0.591$, FID=$89.2$, and DS=$0.532$ which is similar to Tab.1, with no improvement by using a large number of reference images. This indicates that the core problem may not be the data size, and may need specific data curation (including **sample pairs** with only semantic differences in tSA, and similar semantics elsewhere), which poses scalability and applicability challenges.
>
> $ $
>
> **We appreciate the positive feedback and valuable comments of the reviewer. we sincerely hope that reviewers could consider increasing the ratings if our responses have addressed all their questions**.

---

> > ### Comment · Reviewer_89En · 2024-08-09
> >
> > Thank you for clarifying my questions and comments. I agree with the rebuttal that a simple method could be proposed due to thorough analysis, and I believe this paper is indeed a suitable follow-up to ITI-GEN. while I still think it may be incremental because it heavily depends on ITI-GEN method.
> >
> > There is one more thing I’d like to ask. While I also trust that the authors reproduced the exact results since the code is released, I ask for your understanding in confirming the results because the quality of the results in the paper seems to vary significantly. Could you clarify which experiment in this paper achieved similar scores under the exact same settings as ITI-GEN? In the ITI-GEN paper, most results (Table 1 and 2 in their paper) show better or comparable performance in FID compared to the original stable diffusion, but the reproduced results seem to fall short compared to the original model. To be clarifying this, could you also compare the scores with the original model and HP in Table 1 in the proposed paper?
> >
> > Additionally, I have a concern that could be raised not only with this paper but also with the ITI-GEN paper. The poor performance of the HP method might be due to stable diffusion v1.4 not properly reflecting the prompts. Wouldn’t HP perform better with more recent models that use improved text encoders, such as stable diffusion XL or version 3? Is there a possibility to apply this method to these latest models?

---

> ### Author Response · Authors · 2024-08-10
> **Addressing Question on Research Gap and Reproducibility**
>
> Firstly, we really appreciate Reviewer 89En’s prompt and insightful feedback.
>
>
> >Q7: Thank you for clarifying my questions and comments. I agree with the rebuttal that a simple method could be proposed due to thorough analysis, and I believe this paper is indeed a suitable follow-up to ITI-GEN. while I still think it may be incremental because it heavily depends on ITI-GEN method.
>
>
> R7: We would like to emphasize that, as one of our main contributions, we had also proposed new qualitative and quantitative analysis of the effects of learned tokens in individual denoising steps of diffusion models. The proposed method thus grounded on our novel analysis.
>
> $ $
>
> >Q8: which experiment in this paper achieved similar scores under the exact same settings
>
>
> R8: We thank the Reviewer for the question and clarify that our work considers a more fine-grained performance analysis than ITI-Gen which aligns better with existing SOTA literature [27,28,30,31].
>
>
> **Quality evaluation**:  Reproducibility of ITI-Gen’s quality is indeed a question raised by other researchers in ITI-Gen’s GitHub. To address this, ITI-Gen authors clarified (in GitHub issues #1 “Missing Performance Metric” and #7 “The gap between FID indicators is too large”, currently closed) that the FID results in ITI-Gen’s paper Tab.2 are achievable when evaluation is performed on data of all 40 tSAs, for both real and generated images (with learned ITI-GEN prompt) gathered as one large dataset (200*40), and FID is calculated on this large dataset. We have also verified this.
>
> (We will include the links to the GitHub page to AC per NeurIPS guidelines.)
>
> However, as remarked in global responses, such high-level analysis is susceptible to missing the quality degradation existing in specific tSAs (e.g., High-Cheekbone). To address this, in more fine-grain analysis, we instead focus on individual tSA and evaluate their respective quality, aligning with existing fairness literature [27,28,30,31].  With this more fine-grain analysis, we were able to see the distortion in individual learned prompts. In addition, we considered an additional Text-Alignment (TA) quality metric which confirms our findings.
> Note that, Tab. 1 in the original ITI-GEN paper has only fairness comparison for specific tSAs, while FID or other quality evaluation for these tSAs is missing.
>
>
> **Fairness evaluation**: As mentioned in our paper, following existing literature, we utilizes Fairness Discrepancy (FD) metric [27,28,30,31]. Specifically, FD is the same as that implemented with ITI-Gen but instead uses L2 distance in place of  KL divergence.
>
> $ $
>
> >Q9: compare the scores with the original model and HP in Table 1 in the proposed paper?
>
> R9: We provide the following results as per reviewer’s request for HP evaluation:
>
>
> |  | FD | TA | FID  | DS |
> |---|---|---|---|---|
> | Gender | $1.4e^{−3}$ | 0.699 | 77.1 | 0.318 |
> | Young | 0 | 0.674 | 76.4 | 0.392 |
> | Smiling | $8.4e^{−3}$ | 0.672 | 79.8 | 0.323 |
> | High C. | $3.68e^{−3}$ | 0.672 | 80.3 | 0.332 |
> | Pale Skin | $591e^{−3}$ | 0.660 | 96.2 | 0.397 |
> | Eyeglasses | $670e^{−3}$ | 0.670 | 79.1 | 0.468 |
> | Mustache | $554e^{−3}$ | 0.674 | 81.4 | 0.372 |
>
> $ $
>
> As mentioned in our paper e.g., Tab 2, many of the HPs perform poorly w.r.t. Fariness due to their tSA having linguistic ambiguity. However, their sample quality is competitive. Overall, FairQueue still achieves best performance–with competitive fairness as ITI-Gen and high-quality samples as HP.
>
> Due to space limitations, we can provide side-by-side comparison with ITI-Gen and FairQueue in another reply if Reviewer allows us (per NeurIPS rules). (ITI-Gen and FairQueue comparison is already in main paper)
>
> $ $
>
> >Q10: Wouldn’t HP perform better with more recent models that use improved text encoders, such as stable diffusion XL or version 3?  Is there a possibility to apply this method to these latest models?
>
>
> R10: We clarify that we have verified that the problems indicated by the ITI-Gen paper (e.g., in Fig.1) would still persist in more recent versions of Stable Diffusion e.g., SD 3.0. This issue can be observed by simply inputting the prompt “A headshot of a person without glasses” to SD 3.0 demo and generating several samples (provided through the AC comments). The generated samples still frequently have wrong tSA class (“with glasses”) indicating this ambiguity still exists. We provide some results showing the poor fairness of HP with SD3.0. Here, we utilize the same setup as Tab.1 (main paper) with 500 samples per tSA class and HP per Tab.3 (Supp.B.3) to report FD:
>
> $ $
>
> |  | FD |
> |---|---|
> | Eyeglasses | $670e^{−3}$ |
> | Pale Skin | $580e^{−3}$ |
>
> $ $
>
> Regarding applying ITI-Gen or FairQueue to more recent models: while in principle our ideas are model agnostics, it requires some effort to implement in the new SDXL or SD3 code base, beyond the duration of this rebuttal.
>
>
> **Once again, we thank Reviewer 89En’s prompt and insightful feedback.**

---

> > ### Comment · Reviewer_89En · 2024-08-11
> >
> > I see. Now I understand that while the ITI-GEN performance is strong across the entire dataset, there are some attributes where its results suffer from significant degradation, which this paper addresses. Thank you for clarifying my comments. I am raising my score to 5. However, I did not raise it further because I still feel this work is too dependent on the ITI-GEN method, even though it provides valuable improvements for the community.

---

> ### Author Response · Authors · 2024-08-11
> **We appreciate the Reviewer Raising our rating and for the kind words**
>
> We are very thankful for the Reviewer's increased rating and kind words in considering our work “... valuable improvements for the community”.
>
> We respectfully seek the reviewer to consider adjusting their soundness/contributions score to reflect their new understanding, should the reviewer find it appropriate.

---

### Author Rebuttal · Authors · 2024-08-07

Global Response (GR): We thank all the reviewers for their valuable time and effort in reviewing our work. We appreciate the Reviewers' kind comments, such as:

- **Presentation**: Reviewer 89En giving our paper excellent score (4) for presentation; Reviewer iQth for praising our paper as “well-written and logically structured”

- **Analysis**: Reviewer 89En, 8wzu, and 9buQ recognizing our  “highly detailed analysis”, “detailed analysis” and “in-depth analysis”, respectively.

- **Experiments**: Reviewer iQth: recognizing our efforts in making our paper “very rich in experiments”

- **Solution**: Reviewer iQth acknowledging “FairQueue effectively solves the quality degradation problem in ITI_GEN” and Reviewer  9buQ: Acknowledging the “superiority of FaiQueue over ITI-Gen”

$ $

We would also like to express our appreciation to all the Reviewers for allowing us to clarify our work, as well as the constructive comments.  We have considered all comments seriously.

$ $

To briefly recap our work:

1. Our detailed study first uncovers that ITI-Gen –Prompt learning– degrades the quality of the generated samples, an observation missed by recent works [A] –based on high-level analysis.

2. Upon deeper analysis, it is revealed that the quality degradation sources from ITI-Gen’s sub-optimal directional loss used during training, where unrelated knowledge distorts the tokens in ITI-Gen prompt –responsible for encoding the tSA expression.


3. Based on these findings, we deep dive into analyzing the impact of these distorted ITI-Gen tokens during sample generation. Specifically, we inspect the attention maps with our novel H2I/I2H analysis. Our findings reveal to us that:

	- The quality degradation sources from the distorted ITI-Gen tokens affecting the global structure in the early denoising steps

	- Interestingly, given that the global structure is well-synthesized, ITI-Gen tokens work well (in the later stage) to enforce the tSA expression.

4. **Only based on this analysis,** are we then able to propose our Prompt Queuing mechanism (an evolution of H2I), which instead utilizes the Base Prompt in place of HP in the early denoising steps (to first synthesize good global features), followed by ITI-Gen prompt (to enforce tSA expression). Note that HP and Base Prompt share the similarity of being natural language prompts (demonstrated in Supp B.6) -undistorted by the directional loss in contrast to ITI-Gen– and hence produce good global structure.

5. Finally, recognizing that Prompt Queuing would impact the tSA expression due to the reduced denoising steps with ITI-Gen prompt, we propose Attention Amplification which scales up the attention maps for the tSA-specific tokens ($S_i$) resulting in enhanced tSA expression. We coin this combination of Prompt Queuing and Attention Amplification as FairQueue.

6. We then perform comprehensive experiments comparing FairQueue with the existing SOTA ITI-Gen to demonstrate our improved quality and semantic preservation while preserving the fairness performance. Our experiments include:
	-  **3 datasets (fair generative modeling gold-standard benchmarks)**: CelebA [29], FairFace [45], and Fair Benchmark [44]

	- **12 different tSA combinations**

	- **4 popular performance metrics**:  Fairness discrepancy (FD) [16,27,28,30,31], FID [16,38,28,30,31,35,60], Text Alignment [16,37,11,60], DreamSIM [39,B,C].

$ $

Additionally, due to space limitations, we divert some key resources to the supplementary:

1. **Ablation studies** (Supp. A.2): demonstrates the individual contribution of Prompt Queuing and Attention Amplification.

2. **Exploration of ITI-Gen, HP, and Base prompt relationship in their embeddings space** (Supp.B.6): Verified that HP and Base Prompt are similar  (close in their embeddings) and ITI-Gen prompt is dissimilar (distant embeddings).

3. **Hyper Parameters** (Supp. B.3): details for reproducibility.

4. **Code and Qualitative Illustrations** ([Annonymous Link in Supp line 489]).
	- Code and ReadMe
	- Qualitative comparison of ITI-Gen vs FairQueue illustration in \> [more resources] \> [Tab_1_More_illustration]



$ $

In this rebuttal, we have also included a few additional experiments and visualizations, as requested by the reviewers. All of the newly added results help to support the effectiveness of FairQueue:

1.  Human-recognized assessment (Amazon-Mechanical Turk) of the quality and fairness of generated samples by ITI-Gen vs FairQueue

2. Additional Qualitative and quantitative analysis on the attention maps between BasePrompt vs FairQueue vs ITI-Gen

3. Qualitative assessment of the effects of Attention Amplification

4. Evaluating ITI-Gen’s performance with a larger reference dataset (2k)

$ $

In what follows, we provide comprehensive responses to all questions. We could provide more details if there are further questions. We hope that our responses can address the concerns.

$ $

[A] Fernández, Daniel Gallo, et al. "Reproducibility Study of" ITI-GEN: Inclusive Text-to-Image Generation"." Transaction on machine learning research (2024).

[B] Ghazanfari, Sara, et al. "LipSim: A Provably Robust Perceptual Similarity Metric." International Conference on Learning Representations (2024).

[C] Liu, Yinqiu, et al. "Cross-Modal Generative Semantic Communications for Mobile AIGC: Joint Semantic Encoding and Prompt Engineering." arXiv preprint arXiv:2404.13898 (2024).

---

> ### Author Response · Authors · 2024-08-13
> **Updating Reviewers on All Addition Experiments in the Rebuttal**
>
> Once again, we thank all the Reviewers for their insightful comments and for those who considered raising their scores after further clarifications.
>
> To help keep all Reviewers updated, in the following, we provide a summary list of the experiments conducted by us, as suggested by the Reviewers (which we will include in the final manuscript):
>
> $ $
>
> 1. **Assessing the persistence of linguistic ambiguity in a newer version of T2I generative model**: Given that newer versions of the text-to-image models (like SD 3) are using improved text encoders compared to version 1.4, this experiment aims to assess whether these newer versions have a better understanding of prompts with potential linguistic ambiguity e.g., “without eyeglasses”, and therefore whether HP can be used for all/several tSAs in these T2I generative models. Our results show that linguistic ambiguity *still exists in the latest versions* of SD. This aligns with our findings in the main paper (experimented for SD 1.4) showing that HPs are still not viable solutions to address fairness for all tSAs (FD scores for these tSAs still remain large). Therefore, enforcing fairness across all tSAs requires alternative approaches like prompt learning in our proposed FairQueue.
>
> $ $
>
> 2. **User study with human evaluation (Amazon-Mechanical Turk) demonstrating FairQueue’s preferred performance**: In addition to extensive qualitative and quantitative experimental results (using four well-known metrics) already provided in our paper, we have included a user study to compare the quality and fairness of ITI-Gen vs FairQueue based on human assessment. The results show that for all tSAs, the majority of human evaluators also prefer FairQueue sample quality, and the fairness of FairQueue remains competitive with ITI-Gen. These results align with our qualitative and quantitative analysis in the main paper.
>
> $ $
>
> 3. **Additional qualitative and quantitative analysis** on the attention maps between BasePrompt vs FairQueue vs ITI-Gen. Our analysis further demonstrates the effectiveness of FairQueue’s mechanism to first utilize the natural language, Base prompt $T$ to first form a good global structure followed by (tSA aware) learned prompt to synthesize tSA expressions (via localized attention).
>
> $ $
>
> 4. **Qualitative demonstration of the effectiveness of Attention Amplification (AA)**: This is a qualitative ablation study aimed to show how without AA ($c=0$), tSA expression (tSA=’Smiling’) can still be lacking in many samples. Then, with the gradual introduction of AA ($c>0$), the tSA expression is emphasized (samples slowly begin to smile) while maintaining sample quality.
>
> $ $
>
> 5. **Evaluation on a larger reference dataset further supports our proposed solution (2K samples per class of tSA)**: This experiment aims to assess whether the problem of encoding unrelated knowledge can be resolved by increasing the reference dataset size. Our experimental results show that the sub-optimality of the learning objective can not simply be addressed by increasing the reference dataset size, and therefore the quality degradation issue still exists using even larger datasets necessitating using our proposed FairQueue as a solution.
>
> $ $
>
> Overall, all the experiments support FairQueue as an effective solution, achieving both high-quality and fair sample generation, when compared against the existing state-of-the-art.

---

### Decision · Program_Chairs · 2024-09-25

**Decision:**

Accept (poster)

**Comment:**

This work focuses on improving fairness and quality in T2I generation via prompt learning and proposed the FairQueue that is demonstrated to be superior over the current ITI-Gen method. Reviewers are generally positive (one accept and three borderline accept) by acknowledging the contribution and its effectiveness, without obvious objections. Though it mainly targets on improving one specific existing work, extensive experiments verify the advantage of proposed methods. It points out a promising direction of how prompt learning affect the generation and the analysis on attention maps help better understand the black box of big diffusion models, which is of great value to the community. Therefore, after checking all comments, the area chair made a decision of acceptance. Some concerns including adding ablation study and necessary explanations presented in the rebuttal should be addressed in the revised version.